elLIFE

# Interactions between respiratory oscillators in adult rats

Robert TR Huckstepp, Lauren E Henderson, Kathryn P Cardoza, Jack L Feldman*

Department of Neurobiology, David Geffen School of Medicine, University of California, Los Angeles, Los Angeles, United States

**Abstract** Breathing in mammals is hypothesized to result from the interaction of two distinct oscillators: the preBötzinger Complex (preBötC) driving inspiration and the lateral parafacial region ($pF_L$) driving active expiration. To understand the interactions between these oscillators, we independently altered their excitability in spontaneously breathing vagotomized urethane-anesthetized adult rats. Hyperpolarizing preBötC neurons decreased inspiratory activity and initiated active expiration, ultimately progressing to apnea, i.e., cessation of both inspiration and active expiration. Depolarizing $pF_L$ neurons produced active expiration at rest, but not when inspiratory activity was suppressed by hyperpolarizing preBötC neurons. We conclude that in anesthetized adult rats active expiration is driven by the $pF_L$ but requires an additional form of network excitation, i.e., ongoing rhythmic preBötC activity sufficient to drive inspiratory motor output or increased chemosensory drive. The organization of this coupled oscillator system, which is essential for life, may have implications for other neural networks that contain multiple rhythm/pattern generators.

*For correspondence: feldman@g.ucla.edu

**Competing interests:** The authors declare that no competing interests exist.

## Introduction

Coupled oscillator neural networks driving behavior are widespread, e.g., for swimming (*Grillner, 2003*), and locomotion (*Goulding, 2009*; *Talpalar et al., 2013*). Amongst complex and vital behaviors in mammals, breathing, an exceptionally reliable and continuous behavior throughout postnatal life, is one that we may be closest to understanding (*Feldman and Kam, 2015*). Not only do we know the location of the neural microcircuits that generate respiratory rhythm, but we also have direct, accurate and reliable behavioral measures of the output, i.e., breathing. We hypothesize that the respiratory rhythm central pattern generator (CPG) in mammals is comprised of two oscillators (*Feldman et al., 2013*): inspiratory rhythm originates in the preBötzinger Complex (preBötC) in the ventrolateral medulla (*Smith et al., 1991*) and active expiratory rhythm originates in the rostral medulla ventrolaterally adjacent to the facial nucleus (parafacial lateral region; $pF_L$) (*Pagliardini et al., 2011*; *Huckstepp et al., 2015*).

Two parafacial regions that potentially overlap and whose anatomical descriptions are incomplete and ambiguous, the retrotrapezoid nucleus (RTN) and parafacial respiratory group/embryonic parafacial (pFRG/e-PF), contain overlapping subpopulations neurons that express the neurokinin-1 receptor (NK1R), the homeobox gene Phox2b, and the glutamate transporter 2 (VGlut2) (*Nattie and Li, 2002*; *Mulkey et al., 2004*; *Onimaru et al., 2008*; *Thoby-Brisson et al., 2009*; *Ramanantsoa et al., 2011*). Previously, we used unbiased descriptors to partition the ventral and lateral parafacial regions, designating them as parafacial lateral ($pF_L$) and parafacial ventral ($pF_V$). These parafacial regions are chemosensitive (*Mulkey et al., 2004*; *Onimaru et al., 2008*; *Marina et al., 2010*; *Onimaru et al., 2012*), but can be functionally separated by: i) their contribution to active expiration, with the $pF_V$ providing drive to expiration (*Huckstepp et al., 2015*; *Silva et al., 2016*) and the $pF_L$ containing a presumptive expiratory rhythm generator (*Pagliardini et al., 2011*;

**eLife digest** Mammals breathe air into and out of their lungs to absorb oxygen into the body and to remove carbon dioxide. The rhythm of breathing is most likely controlled by two groups of neurons in a part of the brain called the brain stem. One group called the preBötzinger Complex drives breathing in (inspiration), and normally, breathing out (expiration) occurs when the muscles responsible for inspiration relax. The other group of neurons – known as the lateral parafacial region – controls extra muscles that allow us to increase our breathing when we need to, such as during exercise.

Huckstepp et al. set out to determine how these two groups of neurons interact with one another in anesthetized rats to produce a reliable and efficient pattern of breathing. The experiments provide further evidence that inspiration is mainly driven by the preBötzinger Complex. Whilst activity from the lateral parafacial region is needed to cause the rats to breathe out more forcefully than normal, a second low level of activity from another source is also required. This source could either be the preBötzinger Complex, or some unknown neurons that change their activity in response to the levels of oxygen and carbon dioxide in the blood or fluid of the brain.

Further investigation is required to identify how these interactions go awry in diseases that affect breathing, such as sleep apneas.

*Huckstepp et al., 2015*), and; ii) the presence of neurons with respiratory rhythmic behavior in $pF_L$ (or presumptively equivalent areas) (*Onimaru and Homma, 2003*; *Thoby-Brisson et al., 2009*; *Pagliardini et al., 2011*).

In mammals at rest, during wakefulness and sleep, when active breathing movements are primarily inspiratory, generation of the underlying rhythm appears driven by the preBötC. As metabolic demand increases, e.g., during exercise, the $pF_L$ appears to turn on to produce active expiration. Thus, while breathing is a unified act of inspiratory and expiratory airflow, we postulate this behavior results from the coordinated interaction of two anatomically and functionally distinct oscillators (*Mellen et al., 2003*; *Janczewski and Feldman, 2006*; *Pagliardini et al., 2011*; *Huckstepp et al., 2015*). To understand the generation and control of respiration we need to determine how these two oscillators interact. For example, in adult rats the preBötC can generate inspiratory rhythm while the $pF_L$ is quiescent (*Pagliardini et al., 2011*; *Huckstepp et al., 2015*). Is the converse true?

To investigate their independent and interactive functions, we used a pharmacogenetic approach to selectively inhibit the preBötC and/or activate the normally quiescent $pF_L$. We bilaterally transfected the preBötC with the $G_{i/o}$-coupled allatostatin receptors (AlstR), which when activated by allatostatin (Alst) silences transfected preBötC neurons (*Tan et al., 2008*). In the same rats, we bilaterally transfected the $pF_L$ with the $G_q$-coupled $HM_3D$ DREADD receptor ($HM_3DR$) that when activated by clozapine-N-oxide (CNO) depolarizes (*Armbruster et al., 2007*) transfected $pF_L$ neurons. We slowed respiration in a controlled manner by titrating the dose of Alst applied to the AlstR-transfected preBötC, allowing us to examine the dynamics of this presumptive coupled oscillator system as we shifted the balance of activity from the preBötC to the $pF_L$. Depressing preBötC activity resulted in quantal slowing of breathing, similar to slowing of breathing following opiate depression of preBötC activity *in vitro* and in juvenile rats (*Mellen et al., 2003*; *Janczewski and Feldman, 2006*). As preBötC activity waned, burstlets appeared on inspiratory muscle electromyograms (EMGs) and in airflow, consistent with our postulate that burstlets, not bursts, in the preBötC are rhythmogenic (*Kam et al., 2013*). In complementary experiments, we applied CNO to the $HM_3DR$ transfected $pF_L$ to activate this normally quiescent oscillator. We confirmed that $pF_L$ activation initiates active expiration (*Pagliardini et al., 2011*; *Huckstepp et al., 2015*). By combining these protocols, i.e., silencing the preBötC while simultaneously driving the $pF_L$, we removed some confounding factors from these initial experiments. Importantly, we found that active expiration could not be induced when preBötC inspiratory driven motor activity was suppressed and chemosensory drive was absent, indicating that in adult rats active expiration is driven by the $pF_L$ but requires an additional source of network excitation such as ongoing preBötC activity or chemosensory drive. The

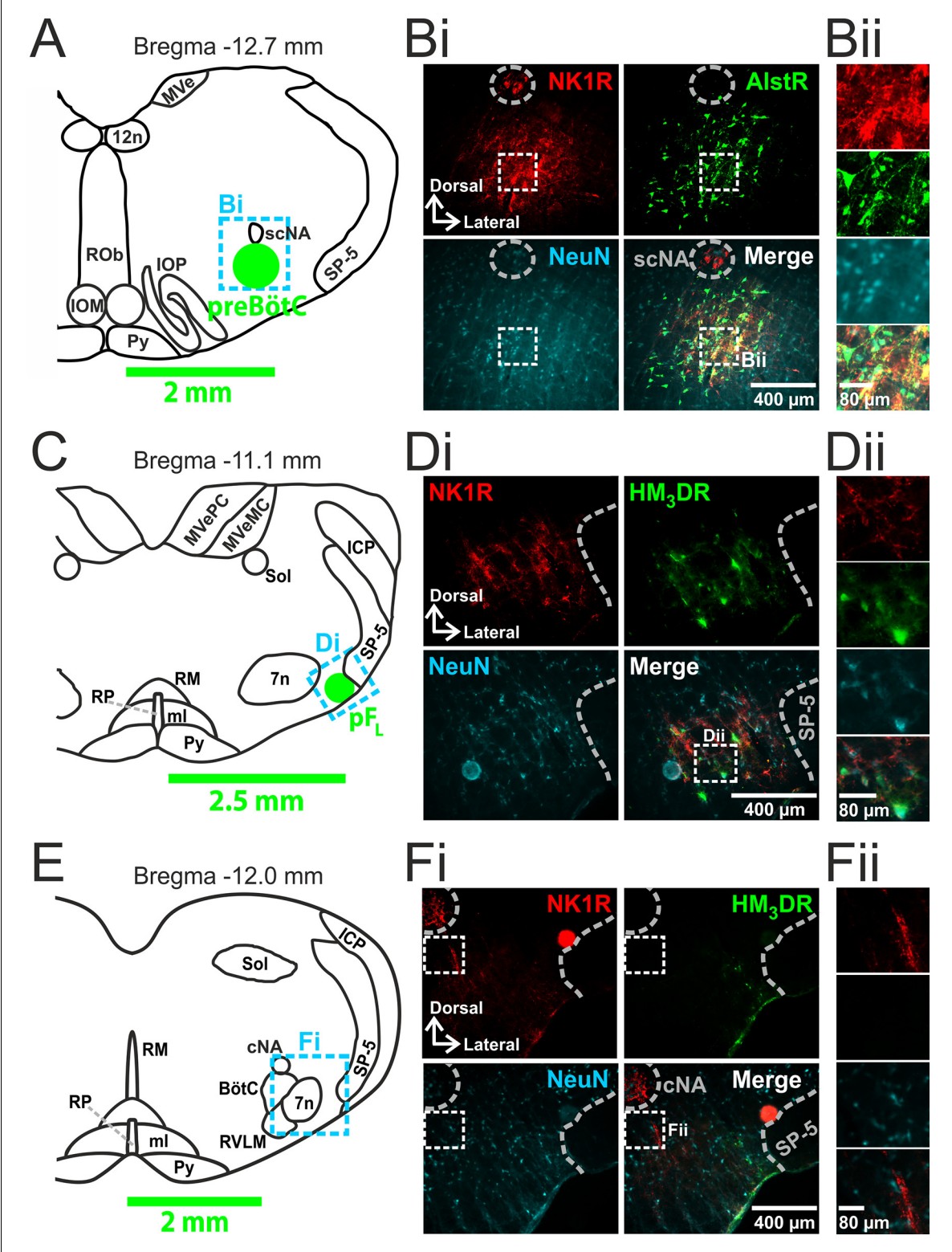

**Figure 1.** Transfection of neurons in preBötC and pF$_L$. (**A**) Localization of preBötC viral injections. Transverse view of medulla at Bregma -12.7 mm, green circle shows location of AlstR-GFP expressing neurons. Dashed blue box indicates location of immunocytochemistry shown in **Bi**. (**B**) Histological analysis of preBötC. (**Bi**) preBötC injection site: neurons identified by NeuN staining (blue) transfected with AlstR expressing GFP (green), co-localized with NK1R (red). (**Bii**) Expanded from **Bi** (dashed white boxes). (**C**) Localization of pF$_L$ viral injections. Transverse view of medulla at Bregma -11.1 mm, green circle shows location of HM$_3$DR-mCitrine expressing neurons. Dashed blue box indicates location of immunocytochemistry shown in **Di**. (**D**)

*Figure 1 continued on next page*

*Figure 1 continued*

Histological analysis of pF$_L$. (**Di**) pF$_L$ injection site: neurons identified by NeuN staining (blue) transfected with HM$_3$D receptor (HM$_3$DR) expressing mCitrine (green), co-localized with NK1R (red). (**Dii**) Expanded micrographs from merged figures in Di (dashed white boxes): NeuN (blue), mCitrine (Green), and NK1R (red). (**E**) Histological analysis of medulla at the level of the Bötzinger Complex (BötC). Transverse view of medulla at Bregma -12.0 mm. Dashed blue box indicates location of immunocytochemistry shown in **Fi**. (**Fi**) No transfected neurons were found in the BötC: neurons identified by NeuN staining (blue) transfected with AlstR or HM$_3$DR co-expressing GFP (green), colocalized with NK1R (red). (**Fii**) Expanded from Fi (dashed white boxes). cNA – compact nucleus ambiguous, scNA – semi-compact nucleus ambiguous, SP-5 – spinal trigeminal tract, 12n – hypoglossal nucleus, 7n – facial nucleus, Py – pyramidal tract, MeV – medial vestibular nucleus, MeVMC – medial vestibular nucleus: magnocellular part, MeVPC - medial vestibular nucleus: parvocellular part, SOL – nucleus of the solitary tract, ROb – raphe obscurus, RP – raphe pallidus, RM – raphe magnus, ml – medial lemniscus, RVLM – rostral ventrolateral medulla, IOM - inferior olive, medial nucleus, IOP - inferior olive principle nucleus, ICP - inferior cerebellar peduncle (restiform body).

organization of this coupled oscillator system may have implications for other neural networks that contain multiple central pattern generators.

## Results

### Viral targeting of the preBötC and pF$_L$

We made two pairs of viral injections in each adult rat, i.e., bilateral injections into the preBötC (*Figure 1A,B*) and into the pF$_L$ (*Figure 1C,D*). In histological sections the preBötC is defined as the neurokinin-1 receptor (NK1R) dense area ventral to the semi-compact nucleus ambiguous (*Figure 1A,B*) and the pF$_L$ is defined as the area ventral to the lateral edge of the facial nucleus, juxtaposed to the spinal trigeminal tract (*Figure 1C,D*) (*Huckstepp et al., 2015*). In representative 40 μm sections: from preBötC injection sites, 161 ± 42, representing 82 ± 6%, neurons expressed GFP (n=3); from pF$_L$ injection sites, 112 ± 31, representing 81 ± 3%, neurons expressed mCitrine (n=3). Transfection sites ranged from ~350–600 μm in diameter. As the responses to Alst and CNO did not differ between the largest and smallest injections sites, the effects of receptor activation were due to silencing or driving of the preBötC and pF$_L$ respectively and not to spread of virus to neighboring regions. We found no labeling of neurons in regions of the medulla other than within the injection sites (data not shown). In particular, we found no fluorescent reporters, i.e., eGFP or mCitrine, in the Bötzinger complex (BötC; *Figure 1E,F*).

### Silencing the preBötC initially slowed breathing frequency (*f*) and induced active expiration, but eventually resulted in apnea

Hyperpolarizing preBötC AlstR-transfected neurons reduced *f*, ultimately progressing to apnea (*Tan et al., 2008*). In anesthetized adult rats at rest transfected with AlstR in the preBötC (n=8; *Figure 2*), Alst injected bilaterally into the preBötC initially decreased *f* (42 IQR 20 to 22 IQR 11 s; p=0.008), increased T$_I$ (0.4 IQR 0.1 to 0.5 IQR 0.2 s; p=0.04) and T$_E$ (1.0 IQR 0.7 to 3.2 IQR 1.0 s; p=0.008), decreased V$_T$ (4.6 IQR 0.7 to 2.4 IQR 0.8 mL; p=0.008) and ∫Dia$_{EMG}$ (25 IQR 13 to 15 IQR 4 arbitrary units (a.u.); p=0.8), and did not alter ∫GG$_{EMG}$ (8.6 IQR 9.8 to 11.1 IQR 13.7 a.u.; p=0.008). When rats became mildly hypercapnic (ETCO$_2$ increased from 39.7 IQR 5.4 to 43.4 IQR 10.6 mm Hg; p=0.008), they exhibited expiratory-modulated ∫Abd$_{EMG}$ (0.3 IQR 0.2 to 9.7 IQR 9.3 a.u., e,g., *Figure 2Aii, Bii*), the signature of active expiration (*Pagliardini et al., 2011*). After inspiratory activity ceased (10.4 IQR 2.7 mins post Alst injection) and rats became severely hypercapnic (ETCO$_2$ 57.8 IQR 22.3 mm Hg), ∫Abd$_{EMG}$ briefly became tonic (17 IQR 5 s.; *Figure 2Aiii, Biii*) before becoming silent, i. e., no activity.

The decrease in *f* was not continuous (n=8; *Figures 2Bii*, *3A–C*), but instead quantal (*Mellen et al., 2003*). In a representative example (*Figure 3C*), kernel density plot estimations determined the optimal bandwidth, i.e., bin size, to be 0.44 s and revealed a multimodal distribution with 3 peaks: control respiratory periods were ~2.1 s, and after Alst injections into the preBötC, respiratory periods increased by quantal multiples of this baseline, i.e., 4.2 and 6.3 s (*Figure 3Ci*). During longer periods, low amplitude activity, which was not present during eupnea, appeared at times when normal breaths were expected (n=8; *Figure 3A,B* red arrows). This low amplitude Dia$_{EMG}$ and airflow activity are postulated to represent the inspiratory motor outflow manifestation of preBötC

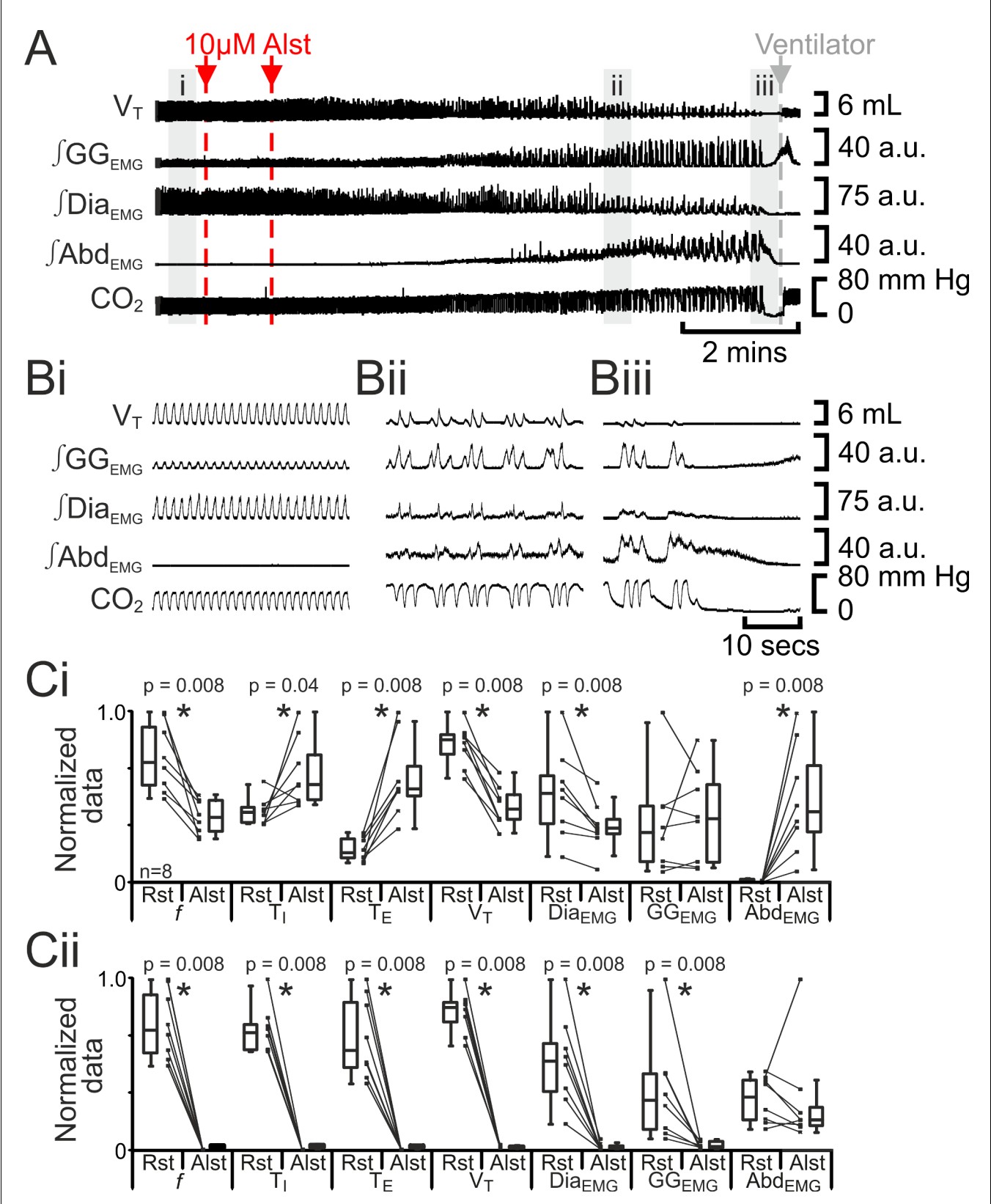

**Figure 2.** Hyperpolarizing preBötC neurons reduced ventilation and induced active expiration, but eventually resulted in apnea. (**A**) Effect of Alst application to left preBötC (unilateral, first red arrow and dashed line) then right preBötC (bilateral, second red arrow and dashed line); gray arrow and

*Figure 2 continued*

dashed line mark onset of mechanical ventilation. (**B**) Expanded traces from A, indicated by shaded epochs (i-iii): (**Bi**) Activity at rest. (**Bii**) and (**Biii**) Activity following bilateral Alst injection. (**C**) Comparison between ventilation in rats at rest (Rst) and following Alst: (**Ci**) before Alst had taken full effect (≈Bii). (**Cii**) After Alst had taken full effect (≈Biii). Lines connect data from individual experiments, box and whisker plots show combined data. Data are normalized to highest parameter, i.e., $f$, $T_I$, $T_E$, $V_T$, $\int GG_{EMG}$, $\int Dia_{EMG}$, or $\int Abd_{EMG}$, value regardless of whether it belonged to control or Alst group. $f$ – frequency, $T_I$ – inspiratory period $T_E$, – expiratory period, $V_T$ – tidal volume, $\int GG_{EMG}$ – integrated genioglossus electromyogram, $\int Dia_{EMG}$ – integrated diaphragm electromyogram, $\int Abd_{EMG}$ – integrated abdominal electromyogram.

burstlets (*Kam et al., 2013*), i.e., low levels of rhythmogenic neural population activity in the pre-BötC that under normal conditions occur in the absence of motor output, being transmitted to motoneuron pools (see Figure 7A in *Kam et al., 2013*). Ultimately, inspiratory activity ceased, i.e., no phasic $\int Dia_{EMG}$ or $\int GG_{EMG}$, and expiratory-related $\int Abd_{EMG}$, i.e., active expiration, terminated, resulting in apnea (n=8; *Figure 2*; p=0.008 for all variables).

We tested for nonspecific effects of Alst injections in non-transfected rats (n=8; *Figure 4*). Injection of Alst into the preBötC did not alter $f$ (47 IQR 12 to 54 IQR 12 s; p=0.1), $T_I$ (0.3 IQR 0.0 to 0.3 IQR 0.0 s; p=0.1), $T_E$ (1.0 IQR 0.3 to 0.8 IQR 0.3 s; p=0.3), $V_T$ (4.7 IQR 1.3 to 4.7 IQR 1.3 mL; p=0.3), $\int Dia_{EMG}$ (34 IQR 13 to 34 IQR 14 a.u.; p=0.5), $\int GG_{EMG}$ (14 IQR 11 to 13 IQR 10 a.u.; p=0.8), nor induce expiratory-related $\int Abd_{EMG}$ (0.3 IQR 0.1 to 0.3 IQR 0.1 a.u.; p=0.3). As there were no effects of Alst in the absence of AlstRs, the cessation of inspiratory activity and termination of expiratory-related $\int Abd_{EMG}$ following injections of Alst into the preBötC of transfected rats resulted from inactivation of AlstR-transfected preBötC neurons.

## Activation of the $pF_L$ leads to active expiration

As we previously found nonspecific effects of CNO at a concentration of 100 μM (*Huckstepp et al., 2015*), we tested for nonspecific effects of 90 μM CNO in non-transfected rats (n=8; *Figure 5*). 90 μM CNO did not alter $f$ (54 IQR 11 to 53 IQR 9 s; p=0.6), $T_I$ (0.3 IQR 0.0 to 0.3 IQR 0.0 s; p=0.2), $T_E$ (0.8 IQR 0.3 to 0.8 IQR 0.2 s; p=0.6), $V_T$ (4.6 IQR 1.5 to 4.7 IQR 1.6 mL; p=0.5), $Dia_{EMG}$ (33 IQR 13 to 34 IQR 14 a.u.; p=1.0), $GG_{EMG}$ (13 IQR 12 to 13 IQR 13 a.u.; p=0.1), nor induce expiratory-related $\int Abd_{EMG}$ (0.3 IQR 0.1 to 0.3 IQR 0.1 a.u.; p=0.3). There were no effects of 90 μM CNO in the absence of $HM_3DRs$.

We predicted that depolarizing $pF_L$ neurons would elicit active expiration, similar to the effect of their disinhibition or optogenetic photoactivation (*Pagliardini et al., 2011*). In anesthetized adult rats at rest transfected with $HM_3DRs$ in the $pF_L$ (n=8; *Figure 6*), CNO (90 μM) increased $f$ (37 IQR 12 to 42 IQR 7 s; p=0.04), did not alter $T_I$ (0.3 IQR 0.1 to 0.3 IQR 0.1 s; p=0.8) decreased $T_E$ (1.3 IQR 0.6 to 1.1 IQR 0.3 s; p=0.02), increased $V_T$ (3.9 IQR 1.3 to 4.2 IQR 1.3 mL; p=0.02), $\int Dia_{EMG}$ (25 IQR 12 to 27 IQR 12 a.u.; p=0.02), and $\int GG_{EMG}$ (7.6 IQR 6.6 to 8.4 IQR 5.6 a.u.; p=0.04), and induced expiratory-related $\int Abd_{EMG}$ (0.3 IQR 0.2 to 3.0 IQR 4.0 a.u.; p=0.02). Thus depolarizing $pF_L$ neurons altered respiration in a similar manner to optogenetic photoactivation (*Pagliardini et al., 2011*) for all measured variables common to both studies. We conclude that depolarization of $HM_3DR$-transfected $pF_L$ neurons resulted in active expiration.

## Silencing the preBötC during active expiration leads to apnea

The preBötC can generate inspiratory rhythm while the $pF_L$ is quiescent (*Pagliardini et al., 2011*). Can the $pF_L$ generate a respiratory rhythm when the preBötC is quiescent? If active expiration is independent of inspiratory activity, then active expiration driven by activation of $HM_3DRs$ in $pF_L$ should persist after cessation of inspiratory activity resulting from injection of Alst in the preBötC. In anesthetized adult rats with an active expiratory breathing pattern induced by CNO, injection of Alst into the preBötC led to a significant decrease in respiratory activity (n=8; p=0.008 for all variables; *Figure 7*), ultimately resulting in apnea, with no rhythmicity in $V_T$, $\int Dia_{EMG}$, $\int GG_{EMG}$, and $\int Abd_{EMG}$. After rhythmic inspiratory activity ceased, $\int Abd_{EMG}$ continued tonically for a short duration (15 IQR 12 s; *Figure 7A,Bii*) before disappearing, which was not different from the duration of tonic $\int Abd_{EMG}$ following the onset of apnea in the absence of CNO (*Figure 2Biii*; p=0.7). When inspiratory activity stopped, rats were severely hypercapnic ($ETCO_2$ 64.6 IQR 14.5 mm Hg; *Figure 7Bii*), this was not different from the hypercapnia at the onset of apnea in the absence of CNO ($ETCO_2$ 57.8

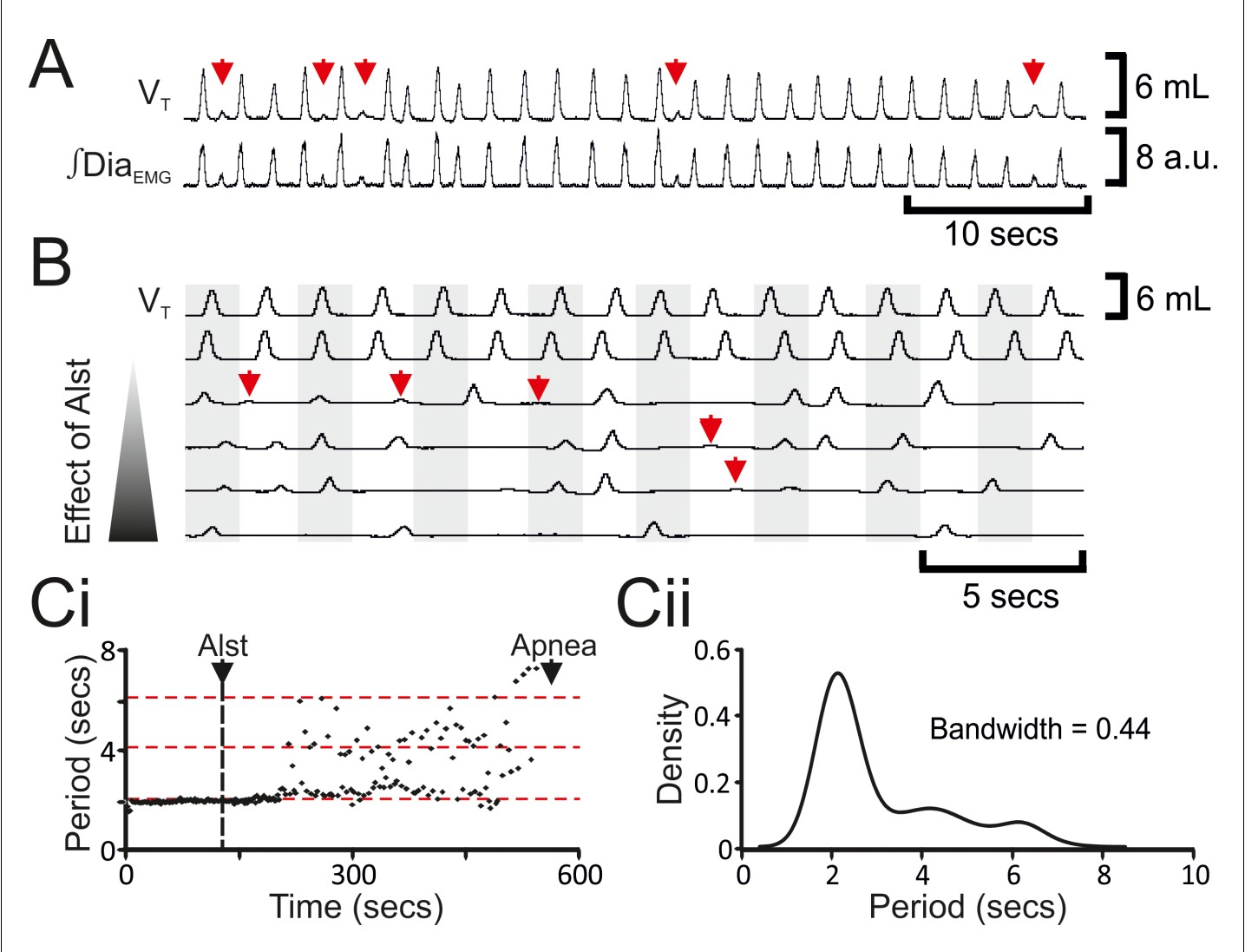

**Figure 3.** Hyperpolarizing preBötC neurons leads to quantal slowing of breathing and burstlet-like activity in $\int Dia_{EMG}$. (A) Burstlet-like activity in airflow and $\int Dia_{EMG}$ traces (red arrows). (B) Traces at different time points (top to bottom ranging from -5 to +10 min) after Alst infusion showing burstlet-like activity. (C) Quantal slowing of breathing. (Ci) Raster plot of respiratory period before and after Alst. (Cii) Kernel density estimations determined the optimal bandwidth, i.e., bin size, of 0.44 s, revealing a multimodal distribution with respiratory periods at quantal intervals of ~2.1 s.

IQR 22.3 mm Hg; p=0.8). Rats were mechanically ventilated to restore $CO_2$ and $O_2$ levels to within the normal range (*Figure 7*). To assess the state of the respiratory CPG the mechanical ventilator was intermittently turned off. Shortly after allatostatin injections into the preBötC, no respiratory activity was seen following cessation of mechanical ventilation, and rats were placed back on the ventilator. As the time from allatostatin injection in the preBötC increased, removal from the ventilator led to brief periods of spontaneous breathing. When spontaneous breathing occurred, the return of inspiratory activity always preceded the return of expiratory abdominal activity; active expiration did not return until $V_T$ reached 3.2 IQR 1.7 mL, $\int Dia_{EMG}$ reached 13.1 IQR 8.2 a.u., and $\int GG_{EMG}$ reached 5.2 IQR 4.2 a.u. (n=8; *Figure 8*). That is, at low levels of reinitiated inspiratory activity there was no active expiration, which appeared only after inspiratory activity reached a higher value. If rats were able to spontaneously breathe without ventilation, they were allowed to do so. However, most early periods of spontaneous breathing deteriorated, and when rats became apneic again they were re-ventilated; in 5 of 6 rats spontaneous breathing occurred during ventilation, and in the remaining rat, spontaneous breathing following removal from the ventilator was sustained. When spontaneous

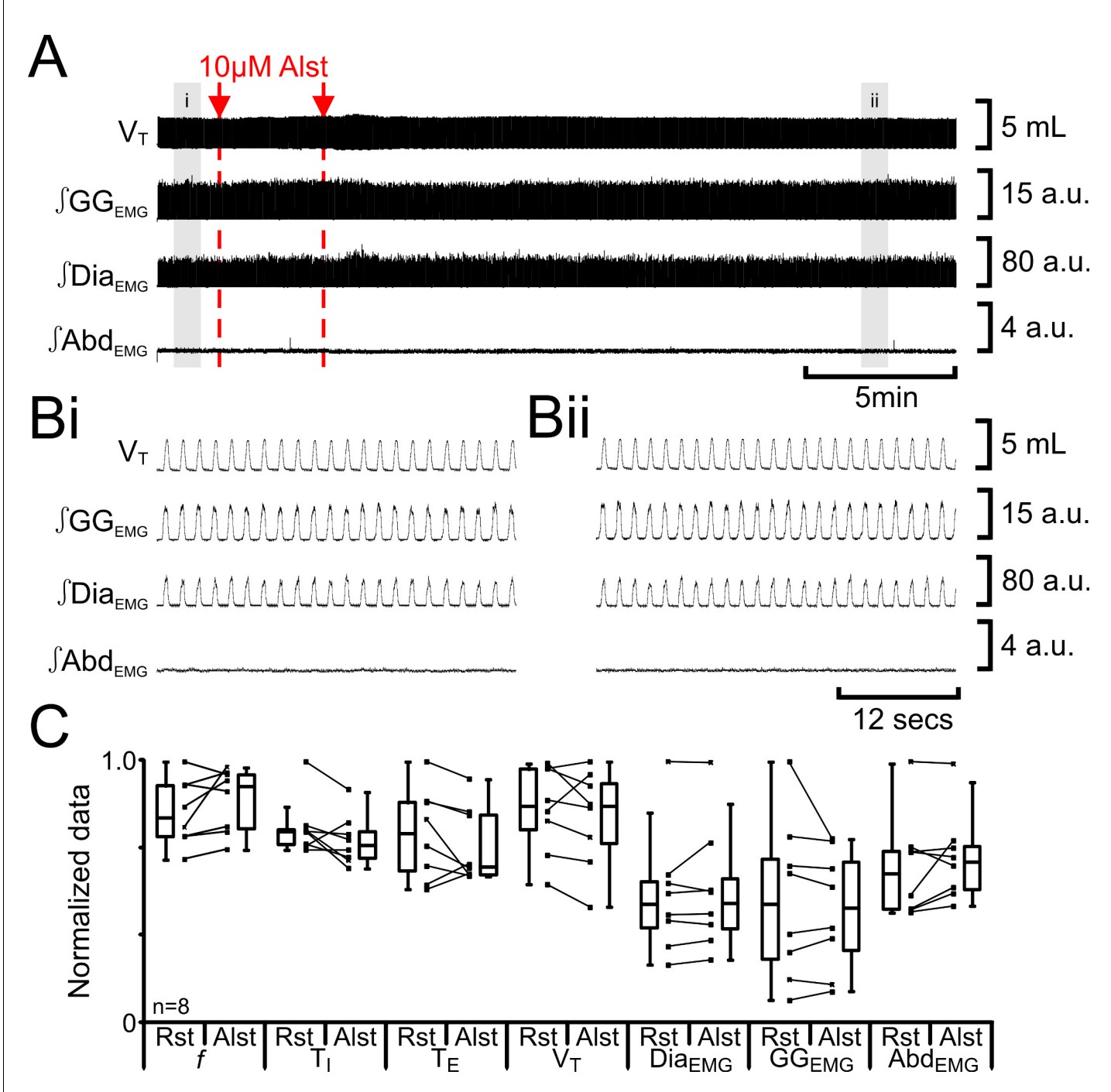

**Figure 4.** Alst in absence of AlstRs does not affect breathing. (**A**) Effect of Alst application to left preBötC (unilateral, first red arrow and dashed line) then right preBötC (bilateral, second red arrow and dashed line). (**B**) Expanded traces from **A**, indicated by shaded epochs (i–ii): (**Bi**) Activity at rest. (**Bii**) Activity following bilateral Alst injection. (**C**) Comparison between ventilation in rats at rest (Rst) and following Alst. Lines connect data from individual experiments, box and whisker plots show combined data. Data are normalized to highest parameter, i.e., f, T_I, T_E, V_T, ∫GG_EMG, ∫Dia_EMG, or ∫Abd_EMG, value regardless of whether it belonged to control or Alst group. Abbreviations defined in *Figure 2*.

breathing occurred during ventilation the return of inspiratory activity always preceded the return of expiratory abdominal activity. Therefore, following silencing of the preBötC with Alst in the presence of CNO to drive the pF_L, at no time during the re-initiation of spontaneous breathing, either with or

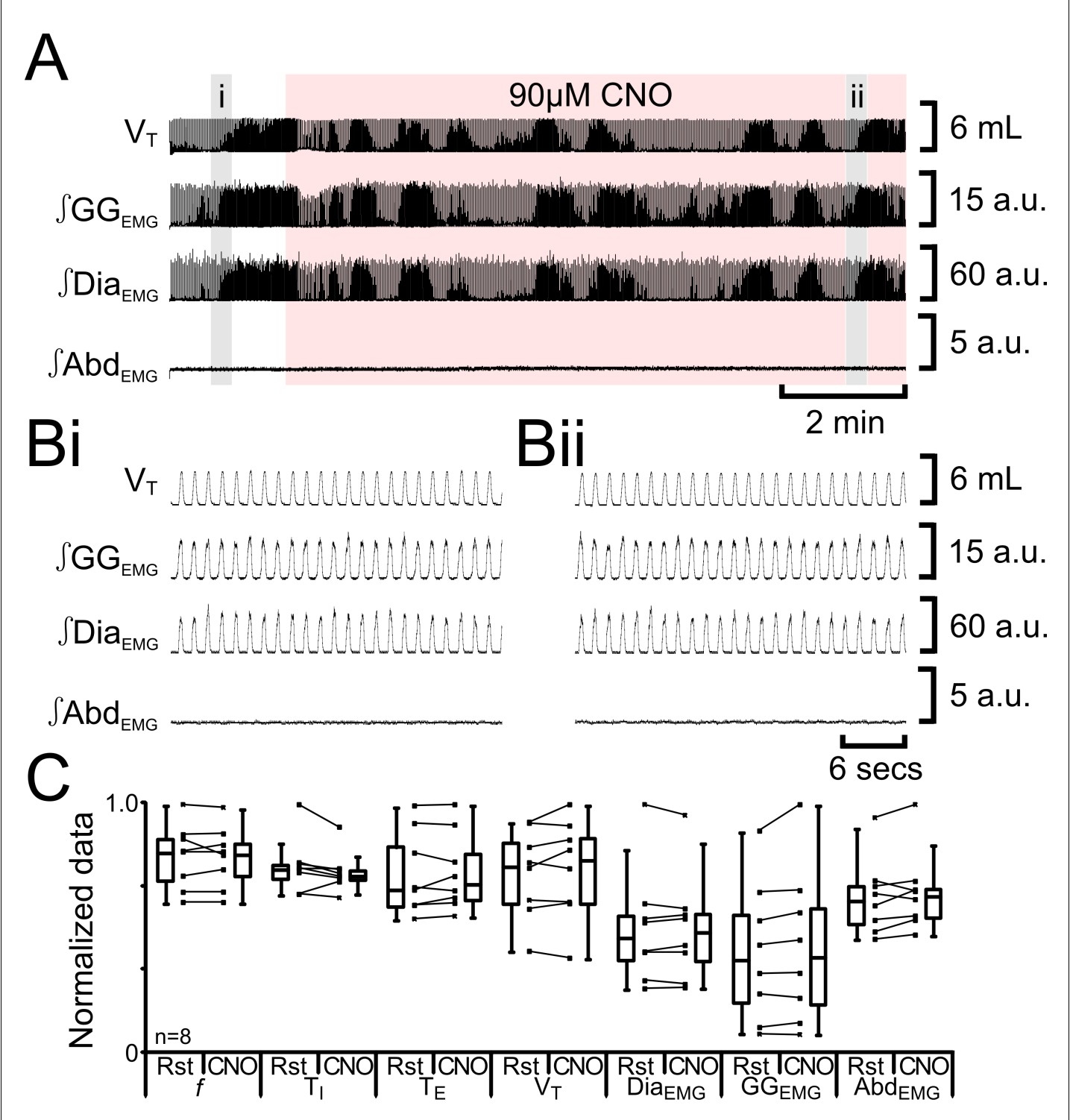

**Figure 5.** CNO in absence of HM₃DRs does not affect respiration. (**A**) Effect of CNO applied to ventral surface (pink shaded area). (**B**) Expanded traces from A, indicated by shaded epochs (i--ii): (**Bi**) Activity at rest. (**Bii**) Activity in presence of CNO. (**C**) Comparison between ventilation in rats at rest and in presence of CNO. Lines connect data from individual experiments, box and whisker plots show combined data. Data are normalized to highest value for that parameter, i.e., $f$, $T_I$, $T_E$, $V_T$, $\int GG_{EMG}$, $\int Dia_{EMG}$, or $\int Abd_{EMG}$ regardless of whether it belonged to control or CNO group. Abbreviations defined in *Figure 2*.

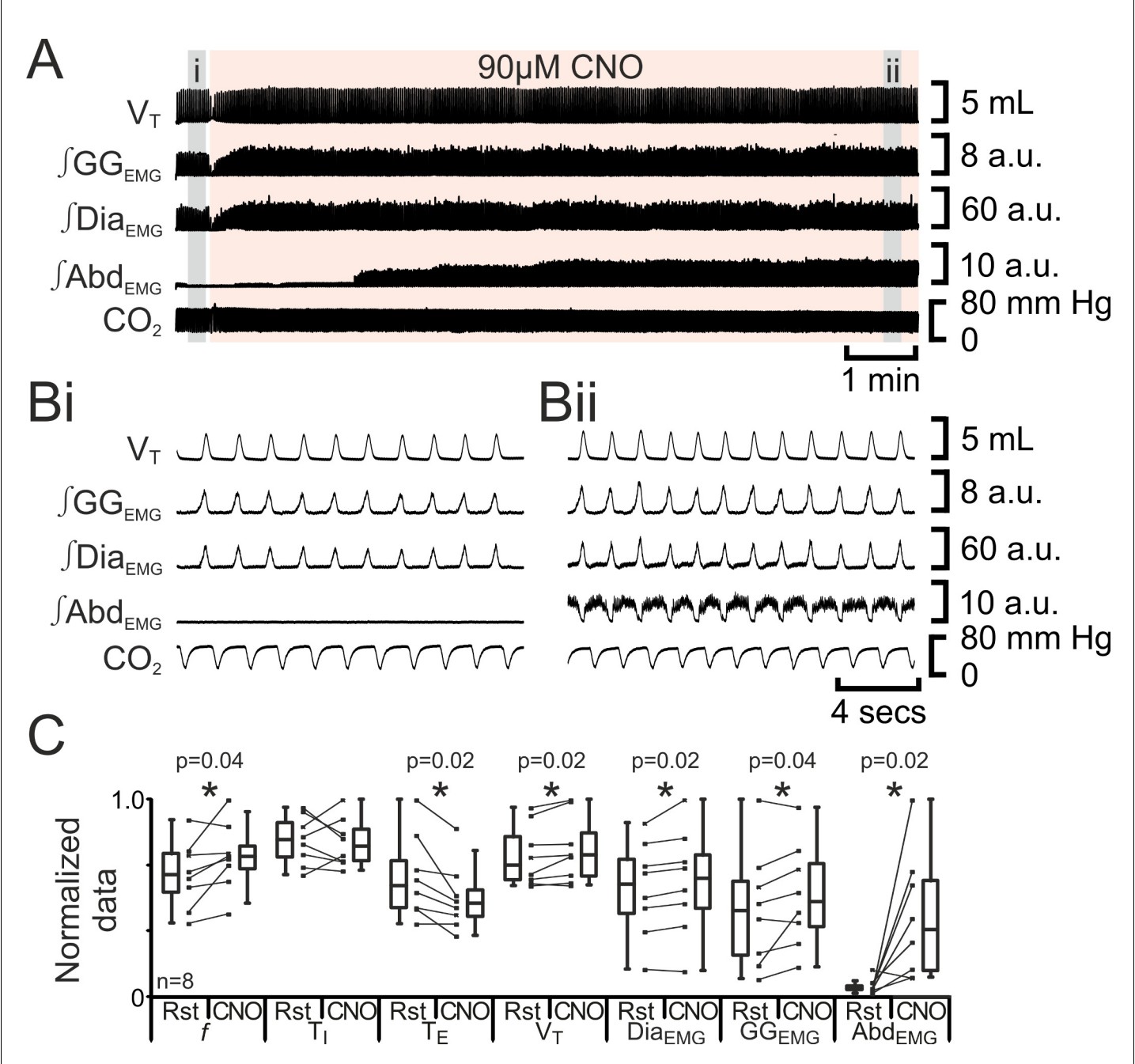

**Figure 6.** Depolarizing $pF_L$ neurons elicits active expiration. (**A**) Effect of CNO applied to ventral surface (pink shaded area). (**B**) Expanded traces from A, indicated by grey shaded epochs (**i-ii**): (**Bi**) Activity at rest. (**Bii**) Activity in presence of CNO. (**C**) Comparison between ventilation in rats at rest and in presence of CNO. Lines connect data from individual experiments, box and whisker plots show combined data. Data are normalized to highest value for that parameter, i.e., $f$, $T_I$, $T_E$, $V_T$, $\int GG_{EMG}$, $\int Dia_{EMG}$, or $\int Abd_{EMG}$ regardless of whether it belonged to control or CNO group. Abbreviations defined in **Figure 2**.

without mechanical ventilation, did active expiration occur in the absence of inspiratory activity and chemosensory drive.

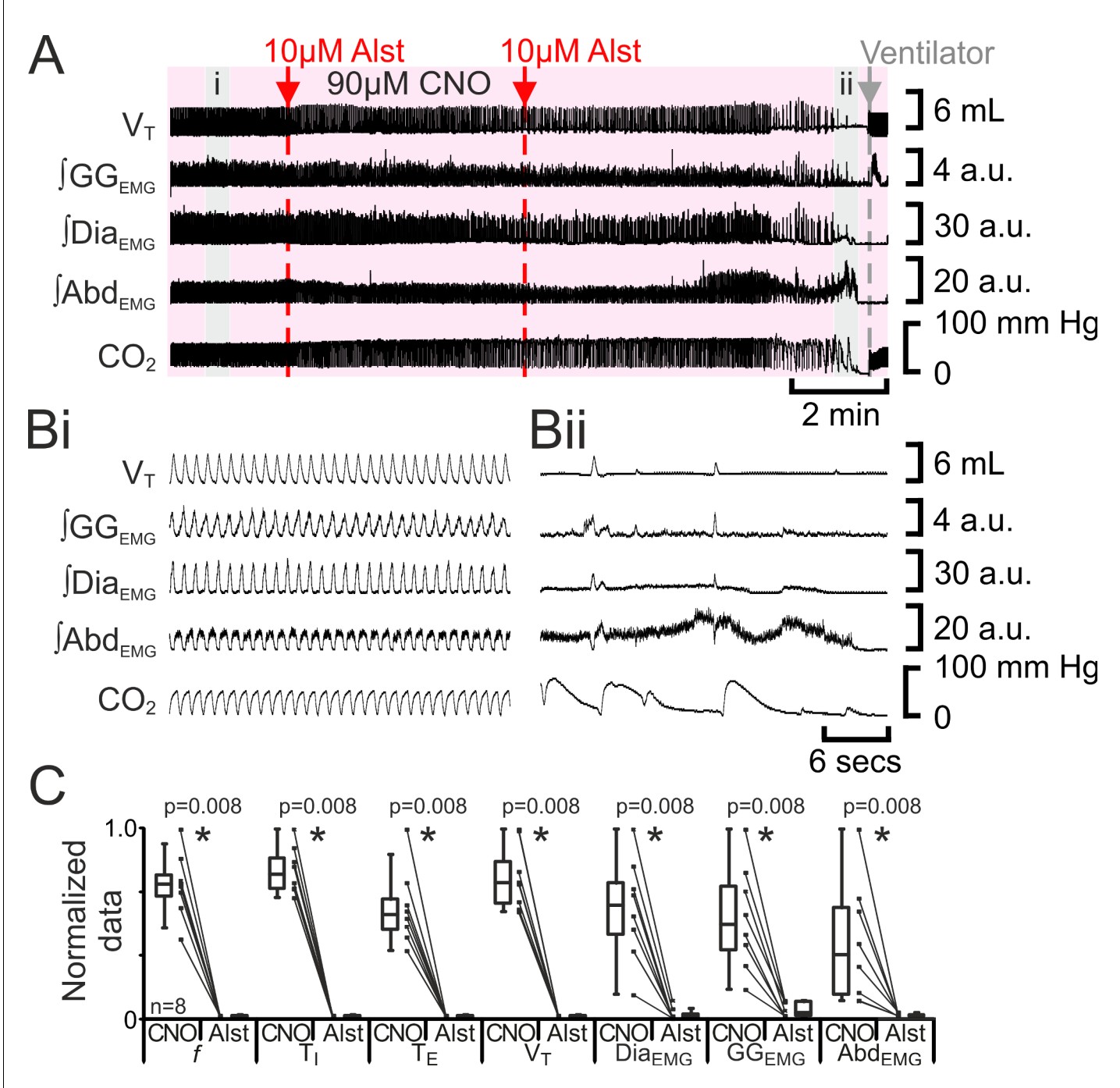

**Figure 7.** Hyperpolarizing preBötC neurons leads to apnea, and loss of active expiration even with activation of pF$_L$. (**A**) Integrated traces from a single experiment showing effect of Alst injection to left preBötC (unilateral, first red arrow and dashed line) then right preBötC (bilateral, second red arrow and dashed line), in presence of CNO (pink shaded area); gray arrow and dashed line mark onset of mechanical ventilation. (**B**) Expanded traces from A indicated by shaded epochs: (**Bi**) In presence of CNO only. (**Bii**) Following Alst in presence of CNO. (**C**) Comparison between ventilation in rats in presence of CNO and following Alst in presence of CNO. Lines connect data from individual experiments, box and whisker plots show combined data. Data are normalized to highest value for that parameter, i.e., $\int GG_{EMG}$, $\int Dia_{EMG}$, or $\int Abd_{EMG}$ regardless of whether it belonged to CNO or CNO with Alst group. Abbreviations defined in *Figure 2*.

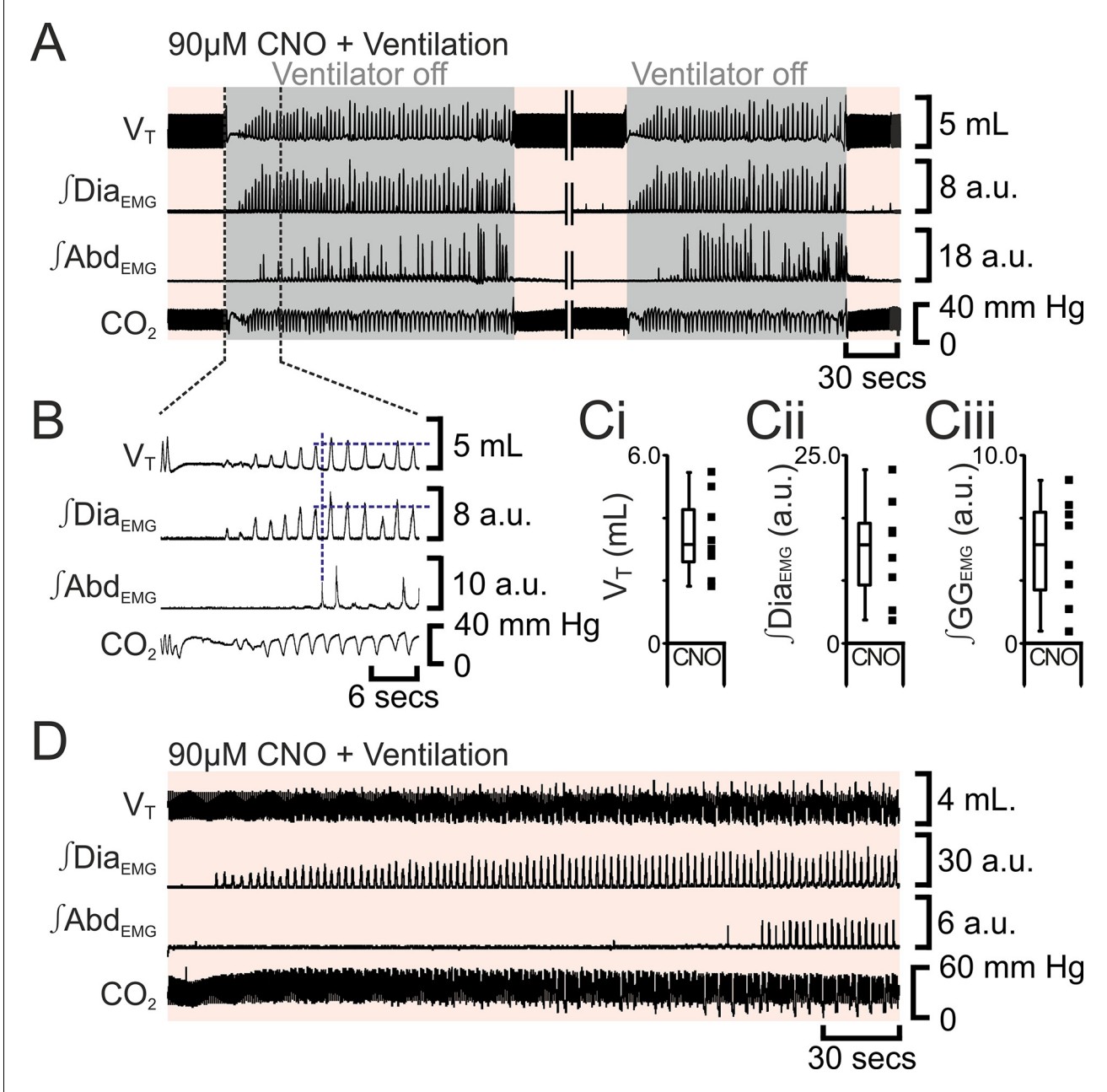

**Figure 8.** Following apnea, in presence of CNO, active expiration only returns after inspiratory activity returns. (A) Integrated traces from one experiment showing effect of removing rat from ventilator (dark grey shaded area), in presence of CNO (pink shaded area), following induction of apnea by microinjection of Alst into the preBötC of preBötC-AlstR transfected rats. Intervening period between the two traces, whilst ventilation was ongoing and continuous, has been removed (double break), so traces can be expanded. (B) Expanded traces from A (indicated by black dashed lines), showing how measurements were taken for inspiratory parameters when active expiration returned. (C) Inspiratory parameters when active expiration returned. Dots represent individual experiments, box and whisker plots show combined data. (Ci) Tidal volume ($V_T$) (Cii) ∫Dia_EMG, (Ciii) ∫GG_EMG. (D) Integrated traces from one experiment showing the return of inspiratory and expiratory activity during mechanical ventilation, in presence of CNO (pink shaded area), following induction of apnea by microinjection of Alst into the preBötC of preBötC-AlstR transfected rats. Abbreviations defined in *Figure 2*.

## Technical consideration

Our experimental design required long lasting activation of receptors at 4 separate loci, i.e., pre-BötC bilaterally and pF_L bilaterally. Optogenetics can have millisecond resolution, making them a

valuable tool for studying breathing on a breath by breath basis, e.g., (*Pagliardini et al., 2011*). However opsins can rapidly (>seconds) desensitize and their activation requires placement of optic fibers to illuminate each transfected area, a particularly challenging problem for four regions in the brainstem. Therefore, we chose a pharmacogenetic approach to excite the pF$_L$ by activation of HM$_3$DRs and inhibit the preBötC by activation of AlstRs. This way we were able to switch neurons in these regions on or off over a period of minutes by application of CNO to the ventral surface of the medulla, and careful titration of the dose and infusion rate of Alst injected into the preBötC.

We note AAV2/5 can retrogradely transfect of some types of afferent neurons, e.g., injections of AAV2/5 into the entorhinal cortex retrogradely labels a subset of dentate gyrus neurons, but not the dense afferent projections from other nuclei (*Aschauer et al., 2013*). We assert that our results are unlikely to be confounded by retrograde transfection of distant neurons: i) Other than the injection site, we found no labeling of medullary neurons following transfection of the pF$_L$ with AAV2/5 viruses that coexpress mCitrine with a DREADD receptor, in this (*Figure 1E,F*) or our previous (*Huckstepp et al., 2015*) study; ii) CNO was applied directly to the ventral medullary surface, and the effective concentration for receptor activation would be limited to the first few hundred microns beneath the ventral surface, and; iii) we find essentially the same effects of activation of neurons in the pF$_L$ as optogenetic photoactivation of the same neurons transfected with lentivirus (*Pagliardini et al., 2011*), which is not retrogradely transported.

Here we discuss the output of the preBötC and pF$_L$, which ultimately form the final output of the respiratory network. Though we do not refer to other nuclei, we do not rule out their contribution to the control of respiration; for example following silencing of the preBötC, the increase in chemosensory drive to the respiratory oscillators, may come from increased drive from other respiratory-related nuclei, such as the pontine nuclei, i.e., locus coreleus, parabrachial nucleus, and the Kolliker Fuse nucleus, the ventral respiratory group or other medullary nuclei, i.e., RTN/pF$_V$, medullary raphe, nucleus tractus solitarii, or even astrocytes. In addition, due to the necessity to perform injections into the medulla, we were unable to record directly from the preBötC or pF$_L$. Therefore, the activity of the preBötC and pF$_L$ was assessed by motor output recorded from respiratory muscles, which may not always reflect the activity of these oscillators, as they may have subthreshold activity, e.g., (*Kam et al., 2013*).

## Discussion

Many rhythmic behaviors are driven by neural networks that (presumably) contain coupled oscillators (*Grillner, 2003*; *Goulding, 2009*). Here, we examined the coupled oscillator microcircuit that controls breathing, a motor behavior that is presently unique insofar as the locations of its oscillators, i.e., pre-BötC and pF$_L$, are known (*Smith et al., 1991*; *Pagliardini et al., 2011*; *Feldman et al., 2013*; *Feldman and Kam, 2015*; *Huckstepp et al., 2015*). In rats, while the function of the preBötC as the critical site for generation of inspiratory rhythm is consistent across all relevant developmental stages, from the third trimester *in utero* through adulthood, and state, e.g., sleep-wake, rest-exercise (*Smith et al., 1991*; *Janczewski and Feldman, 2006*; *Tan et al., 2008*; *Kam et al., 2013*), the presumptive function of the pF$_L$ and its coupling with the preBötC appears to change developmentally and with state (*Figure 9*) (*Onimaru and Homma, 2003*; *Iizuka and Fregosi, 2007*; *Oku et al., 2007*; *Onimaru et al., 2008*; *Thoby-Brisson et al., 2009*; *Pagliardini et al., 2011*; *Huckstepp et al., 2015*). To understand the context in which we interpret our data, we first summarize this disparate literature with reference to *Figure 9*.

From third trimester *in utero* to adulthood, NK1R-expressing neurons lying under the lateral edge of the facial nucleus and extending laterally out to the spinal trigeminal tract have respiratory-rhythmic burst activity (*Onimaru et al., 2008*; *Thoby-Brisson et al., 2009*; *Pagliardini et al., 2011*; *Huckstepp et al., 2015*). In E14.5 mice, these neurons are designated as comprising the embryonic parafacial nucleus (e-pF) (*Thoby-Brisson et al., 2009*); in postnatal rats, these neurons are designated as comprising the parafacial respiratory group (pFRG) (*Onimaru and Homma, 2003*), and; in the adult rat, these neurons are designated as comprising the pF$_L$ (*Pagliardini et al., 2011*; *Huckstepp et al., 2015*). While 3 distinctly different groups of NK1R neurons expressing respiratory-rhythmic bursting could exist in sequence within this homologous anatomical location at distinctly different developmental time points, the most parsimonious interpretation is of one population of

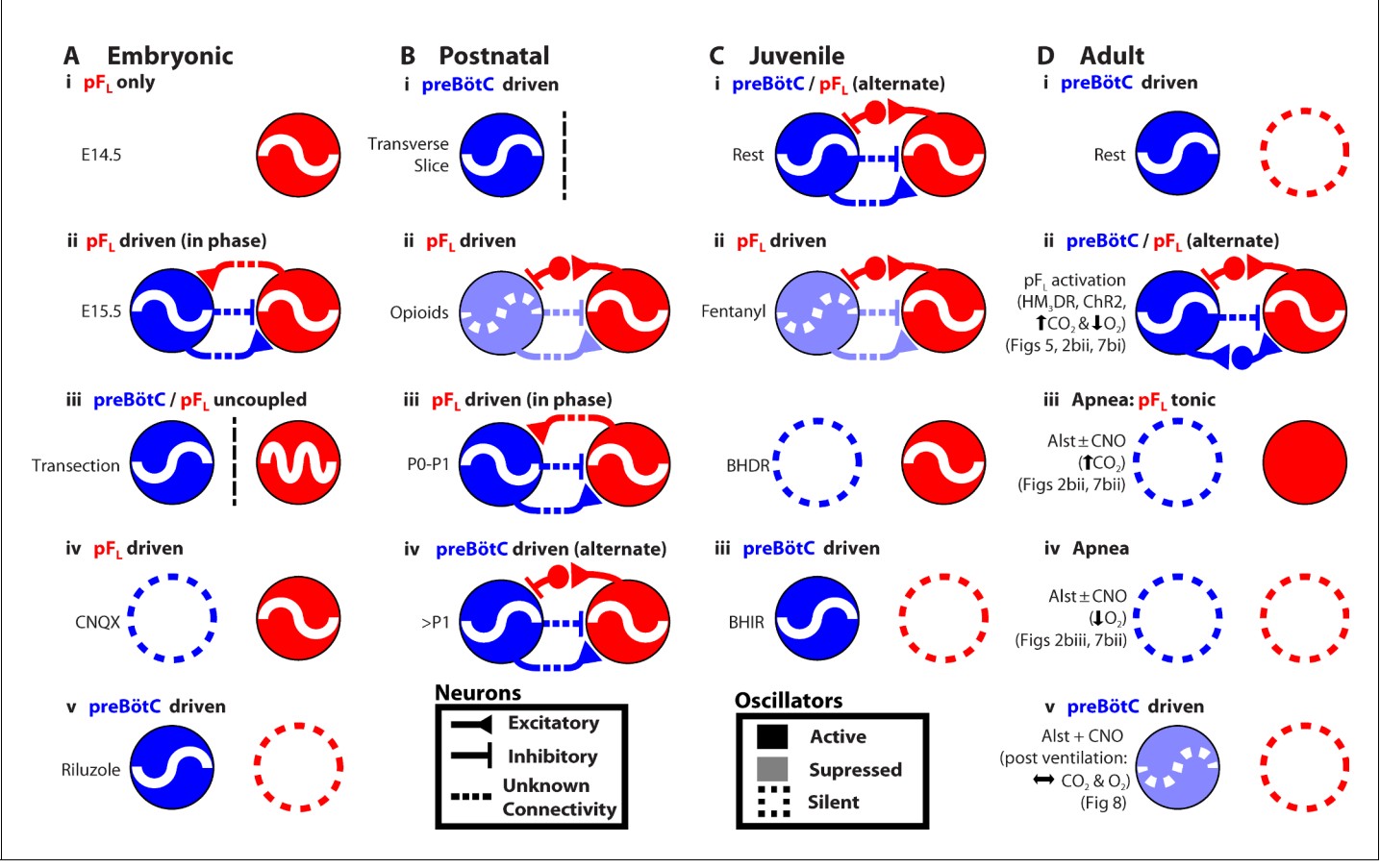

**Figure 9.** Developmental and state-dependent changes in coupling between pF_L and preBötC. Functional connections of undetermined connectivity are indicated as broken lines. As pF_L neurons are excitatory (*Onimaru et al., 2008*; *Thoby-Brisson et al., 2009*) and lack inhibitory markers (*Ellenberger, 1999*; *Tanaka et al., 2003*), inhibitory connections from pF_L to preBötC are indirect (see *Figure 7* in *Huckstepp et al., 2015*). (**A**) Embryonic stage (all data *in vitro*). (**Ai**) pF_L (red circle) oscillates at embryonic day 14.5 (E14.5). (**Aii**) preBötC (blue circle) oscillates at embryonic day 15.5 (E15.5) and couples to the pF_L, where it excites and inhibits different groups of pF_L neurons. (**Aiii**) preBötC and pF_L can oscillate independently of each other following a transverse section caudal to facial nucleus. (**Aiv**) pF_L can oscillate in the absence of preBötC following bath application of a glutamatergic antagonist (CNQX). (**Av**) preBötC can oscillate in the absence of the pF_L following bath application of a sodium channel blocker (riluzole) (*Thoby-Brisson et al., 2009*). (**B**) Postnatal stage. (**Bi**) In late fetal (*Thoby-Brisson et al., 2005*; *Bouvier et al., 2008*) and postnatal rats (*Smith et al., 1991*), the preBötC can oscillate in the absence of the pF_L in transverse slices, and (**Bii**) the pF_L can oscillate independently following suppression of preBötC rhythm by bath application of opioid agonists (*Takeda et al., 2001*; *Janczewski et al., 2002*). Biii) Immediately following birth, respiratory rhythm is driven by pF_L (*Onimaru and Homma, 2003*; *Oku et al., 2007*). (**Biv**) Shortly after birth (>1 day), the breathing CPG becomes driven by the preBötC (*Oku et al., 2007*). (**C**) Juvenile stage. (**Ci**) Expiration and inspiration alternate and are reciprocally coupled. (**Cii**) PreBötC and pF_L are differentially affected by fentanyl, which shifts breathing to an expiratory-dominant pattern. (Cii + iii) preBötC and pF_L can be independently suppressed by activation of Breuer-Hering deflation reflex (BHDR; **Cii**) or inflation reflex (BHIR; **Ciii**) (*Janczewski and Feldman, 2006*). (**D**) Adult Stage: (**Di**) breathing is inspiratory driven by preBötC while pF_L activity is normally suppressed at rest (also see *Pagliardini et al., 2011* and *Huckstepp et al., 2015*); (**Dii**) activation of HM_4DR transfected pF_L neurons by CNO (see *Figure 6*) or optogenetic activation (*Pagliardini et al., 2011*), or suppression of AlstR transfected preBötC neurons with Alst (see *Figure 2*, *7*) can induce active expiration; (**Dii**) as preBötC neurons project to the pF_V but do not appear to project to the pF_L (*Tan et al., 2010*), excitatory drive from the preBötC to the pF_L is most likely through an intermediate excitatory relay, such as the pF_V. (**Diii**) Depression of inspiration by Alst, in presence or absence of CNO, leads to tonic expiratory activity during hypercapnia (see *Figures 2*, *7*) or (**Div**) apnea during hypoxia (see *Figures 2*, *7*). (**Dv**) As breathing returns, abdominal activity remains absent until inspiratory activity is near normal levels (see *Figure 8*), implicating an indirect involvement of preBötC excitatory neurons in expiration either through its excitatory projections throughout breathing CPG (*Tan et al., 2010*), including pF_V that contributes to expiratory activity (*Huckstepp et al., 2015*), or through mechanosensory feedback that can provides expiratory drive (*Remmers, 1973*; *Davies and Roumy, 1986*; *Janczewski and Feldman, 2006*).

neurons studied at different developmental time points; we postulate that this is the case. For the balance of the DISCUSSION, we will refer to this location as the pF_L.

NK1R-expressing neurons in the ventral respiratory column (including the parafacial region) are almost exclusively glutamatergic (*Guyenet et al., 2002*). In embryonic and postnatal rodents, the

pF$_L$ is comprised of glutamatergic NK1R-expressing neurons (*Onimaru et al., 2008*; *Thoby-Brisson et al., 2009*). In the adult rodent, the presumptive pF$_L$ expiratory oscillator is comprised (mostly) of NK1R-expressing neurons (*Figure 1D*, see also *Huckstepp et al., 2015*). Since the pF$_L$ and pF$_V$ do not appear to contain any inhibitory neurons (*Ellenberger, 1999*; *Stornetta and Guyenet, 1999*; *Tanaka et al., 2003*; *Fortuna et al., 2008*; *Abbott et al., 2009*), we conclude that any resultant inhibitory action associated with their activity occurs through an intermediate relay of inhibitory neurons, perhaps located in the preBötC (*Morgado-Valle et al., 2010*) or BötC (*Schreihofer et al., 1999*).

In E14.5 mice, the pF$_L$ is rhythmic before the preBötC appears to form (*Figure 9Ai*; see also *Thoby-Brisson et al., 2009*). At E15.5, the preBötC forms (*Thoby-Brisson et al., 2005*) and it becomes rhythmically coupled to the pF$_L$ (*Thoby-Brisson et al., 2009*). At this time: i) the preBötC and pF$_L$ can oscillate independently following a transverse section caudal to facial nucleus (*Figure 9Aiii*); ii) the pF$_L$ can oscillate in the absence of preBötC rhythm following bath application of the glutamatergic antagonist CNQX (*Figure 9Aiv*), and; iii) the preBötC can oscillate in the absence of the pF$_L$ following administration of the sodium channel blocker riluzole (*Figure 9Av*) (*Thoby-Brisson et al., 2009*). Thus embryonically, the preBötC and pF$_L$ are independent, coupled oscillators, and rhythmic output of the network is a convolution of the faster rhythm of the pF$_L$ and the slower rhythm of the preBötC (*Thoby-Brisson et al., 2009*); the pF$_L$ (*Thoby-Brisson et al., 2009*) provides excitatory drive to the preBötC, while bursting in the preBötC, comprised of inhibitory (*Rahman et al., 2015*) and excitatory neuronal activity (*Gray et al., 2010*), inhibits and excites pF$_L$ neurons (*Thoby-Brisson et al., 2009*). Whether the projections between the preBötC and pF$_L$ are direct or indirect (*Figure 9Aii*) is yet to be determined (*Thoby-Brisson et al., 2009*).

In postnatal rats, respiratory rhythm appears similar to the embryo; the glutamatergic pF$_L$ provides excitatory drive to the preBötC (*Onimaru and Homma, 2003*; *Onimaru et al., 2012*), while bursting of the preBötC, comprised of inhibitory (*Morgado-Valle et al., 2010*) and excitatory neurons (*Gray et al., 2010*), inhibits and excites different subsets of pF$_L$ neurons (*Takeda et al., 2001*; *Onimaru et al., 2007*). At this time: i) the preBötC can oscillate in transverse slices, i.e., in the absence of the pF$_L$ (*Figure 9Bi*) (*Smith et al., 1991*) and; ii) the pF$_L$ can oscillate following suppression of preBötC rhythm by bath application of opioid agonists (*Figure 9Bii*) (*Takeda et al., 2001*; *Janczewski et al., 2002*). Thus postnatally, the preBötC and pF$_L$ are also independent, coupled oscillators. Immediately following birth, respiratory rhythm appears driven by the pF$_L$ (*Figure 9Biii*) (*Onimaru and Homma, 2003*; *Oku et al., 2007*), perhaps to protect against the opioid surge in the fetal brain during birth (*Janczewski et al., 2002*) that would act to suppress preBötC activity (*Takeda et al., 2001*). Shortly after birth (~P2-P4), the breathing central pattern generator (CPG) matures, and becomes driven by the preBötC (*Oku et al., 2007*; *Mckay et al., 2009*; *Kennedy, 2015*) (*Figure 9Biv*), and remains so throughout life.

In juvenile rats, the preBötC and pF$_L$ are reciprocally coupled (9Ci) but are still differentially affected by fentanyl, which shifts breathing to an expiratory-dominant pattern (9Cii) (*Janczewski and Feldman, 2006*). Also, the preBötC and pF$_L$ can be independently suppressed by the Breuer-Hering deflation reflex (BHDR; 9Cii) or inflation reflex (BHIR; 9Ciii) (*Janczewski and Feldman, 2006*).

Here, we used a novel pharmacogenetic strategy to investigate how the preBötC and pF$_L$ interact to produce the appropriate breathing pattern in adult spontaneously breathing vagotomized rats (*Figure 9D*). The preBötC is the dominant oscillator, active at rest to drive inspiratory movements (*Figure 9Di*, see also *Pagliardini et al., 2011* and *Huckstepp et al., 2015*). The pF$_L$ is the subsidiary conditional oscillator, quiescent at rest (*Figure 9Di*); in the presence of concurrent, or reduced, preBötC activity (*Figure 2*), and appropriate inputs, such as an increase in expiratory drive through activation of exogenous receptors (*Figure 6*, see also *Pagliardini et al., 2011*) or altered blood gases (*Huckstepp et al., 2015*), the pF$_L$ drives active expiration (*Figure 9Dii*). In the absence of preBötC activity, chemosensory drive increases network excitability and drives active expiration, though abdominal activity is tonic due to a lack of phasic inhibition from the preBötC (*Figure 9Diii*). However, the pF$_L$ appears incapable of independently driving respiratory movements in the adult rat when preBötC activity is silenced by exogenous receptors (*Figures 2, 7, 9Div*) or when preBötC activity is low and blood gases are normal (*Figures 8, 9Dv*). As preBötC neurons project to the pF$_V$ but do not appear to project to the pF$_L$ (*Tan et al., 2010*), excitatory drive from the preBötC to the pF$_L$ is most likely through an intermediate excitatory relay, such as the pF$_V$ (*Figure 9Dii*).

## Silencing the preBötC leads to quantal slowing of breathing and transmission of burstlets to motoneuron pools

Hyperpolarizing preBötC neurons initially decreased $f$, $V_T$, and $\int Dia_{EMG}$ (*Figures 2*, *3*, see also *Tan et al., 2008*). Breathing slowed in a quantal manner (*Mellen et al., 2003*) with missed inspiratory bursts, rather than through a gradual increase in period (*Figures 2Bii*, *3A–C*). Concurrently, as drive within the preBötC diminished and chemosensory drive increased, presumptive burstlets in the pre-BötC (*Kam et al., 2013*) appeared to be transmitted to motor pools, leading to low level inspiratory motor activity, i.e., $\int Dia_{EMG}$ and $\int GG_{EMG}$, resulting in minimal inspiratory airflow (*Figure 3A–B*). Here, lowered drive in the preBötC from activation of the allatostatin receptor creates burstlets in the preBötC; under normal conditions these burstlet signals are not strong enough to drive motor activity. However, increased excitability in the premotor and motor network caused by increased chemosensory drive (from decreased ventilation) causes these normally subthreshold events to become suprathreshold, and thus these burstlets are transmitted to the motor output. During quantal slowing of breathing, inspiratory activity on the $Dia_{EMG}$ and $GG_{EMG}$ was interspersed with $Abd_{EMG}$ activity (*Figures 2A-B*, *7A–B*). Interestingly, even when expiratory activity was at its highest, low level preBötC activity, as seen as burstlets on the $Dia_{EMG}$ and $GG_{EMG}$ was still able to inhibit $Abd_{EMG}$ activity (*Figure 7Bii*). These observations are consistent with our hypothesis that burstlets originate in the preBötC and, under atypical conditions can be transmitted to motoneuron pools to be seen as small events in muscle EMGs (*Kam et al., 2013*).

## Silencing the preBötC leads first to active expiration then to apnea

Following hyperpolarization of preBötC neurons by activation of AlstRs, active expiration appeared (*Figures 2*, *3*, *9Dii*). Hypoxia and hypercapnia can themselves induce active expiration (*Iizuka and Fregosi, 2007*; *Huckstepp et al., 2015*), whether changes in blood gases are sensed by $pF_L$ neurons, or by neurons driving the $pF_L$, remains to be determined. In either case, active expiration could have been due to increased hypoxic and/or hypercapnic drive to the $pF_L$ as inspiratory movements waned (*Figures 2A–B*, *7A–B*, *9Dii*). Alternatively, active expiration could be due to disinhibition of a conditional expiratory oscillator resulting from a loss of (presumptive) inhibitory preBötC drive (*Kuwana et al., 2006*; *Morgado-Valle et al., 2010*) to the $pF_L$ (*Figure 9Dii*). Subsequently, when inspiratory motor outflow disappeared and hypercapnia and hypoxia inexorably increased, active expiration ceased (*Figures 2*, *9Div*, see also *Tan et al., 2008*), perhaps due to the inhibitory effect of severe hypoxia on expiratory motor output (*Sears et al., 1982*; *Fregosi et al., 1987*). At normal blood gas levels, active expiration may be seen in the absence of inspiratory motor activity resulting from lung inflation, e.g., in juvenile rat (*Janczewski and Feldman, 2006*) (*Figure 9C*). Nonetheless, from the data presented here in the adult rat, we conclude a necessary role of preBötC excitatory drive in generating active expiration in normal breathing (discussed further below).

## Activating the $pF_L$ can induce active expiration but not during to apnea

Depolarizing $pF_L$ neurons by activation of $HM_3DRs$ (*Figure 6*) changes breathing in a manner similar to their disinhibition or optogenetic photostimulation (*Pagliardini et al., 2011*). In the presence of inspiratory motor output, depolarizing $pF_L$ neurons led to substantial active expiration, but when inspiration ceased so did active expiration (*Figures 7*, *9Div*). Tupal *et al* observed the absence of expiratory rhythm in perinatal mice lacking Dbx1 neurons and concluded that Dbx1-derived parafacial neurons are an essential component of an expiratory oscillator (*Tupal et al., 2014*). Our data suggests an alternative interpretation, that suppressing preBötC neuron activity, either acutely as done here or perhaps genetically via Dbx1 deletion, is sufficient to prevent active expiration, without any need to invoke explicit perturbations of the $pF_L$.

## Loss of inhibitory output from the preBötC leads to a loss of phasic expiratory activity

As rats went from eupnea to apnea following suppression of preBötC activity, and after initiation of active expiration, $\int Abd_{EMG}$ transitioned from phasic to tonic: $\int Abd_{EMG}$ oscillated with a normal anti-phase relationship to inspiration when preBötC drive and hypercapnia were moderate, but became tonic during apnea accompanied by severe hypercapnia (*Figures 2A–B*, *7A–B*, *9Diii*). Similarly, in anesthetized cats as ventilation moves in the opposite direction from hypocapnic apnea to eupnea

with a resultant increase in $CO_2$, expiratory motor activity transitions from tonic to phasic (*Sears et al., 1982*). We suggest, like Sears *et al*, that near the transition from eupnea to apnea, expiratory activity is phasic due to periodic inhibition of tonic activity during inspiration, and becomes tonic once the phasic inspiratory inhibition is lost (*Figures 2Bii–iii*, *7Bii*, *9Diii*). We suggest the loss of phasic inhibition is the result of silencing of inhibitory preBötC inspiratory neurons by Alst (*Kuwana et al., 2006*; *Morgado-Valle et al., 2010*), implying a significant role for inhibitory preBötC neurons in shaping expiratory output.

## Respiratory network activity is a requirement for active expiration

Interestingly, tonic $\int ABD_{EMG}$ activity was not seen during the transition from apnea to eupnea (*Figures 8Aii*, *9Dv*) in contrast to its presence during the reverse transition from eupnea to apnea (*Figures 7Bii*, *9Diii*). Following apnea during ventilation to maintain blood gases, when $pF_L$ neurons were excited by activation of $HM_3DRs$ there were three phases to the re-initiation of breathing. Initially, soon after silencing the preBötC, no inspiratory or expiratory activity was seen, even in the absence of mechanical ventilation. Next, presumably as the effect of Alst on preBötC neurons was waning, no inspiratory or expiratory activity was seen during ventilation, but spontaneous breathing was present upon removal from the ventilator. Here, spontaneous breathing was likely due to increased chemosensory drive to the preBötC overcoming the waning hyperpolarizing effect of the Alst on preBötC neurons. Ultimately, as spontaneous breathing continued and chemosensory drive diminished, rats would once again become apneic and require mechanical ventilation (*Figure 8A–C*). During this phase, upon removal from the ventilator, $\int Abd_{EMG}$ did not return until inspiratory motor, and presumably preBötC, activity reached a threshold level (*Figure 8A–C*). Finally, spontaneous breathing occurred during ventilation, once the effect of Alst on preBötC neurons had worn off (*Figure 8D*). During this phase, inspiratory activity on the $GG_{EMG}$ and $Dia_{EMG}$ always returned before $Abd_{EMG}$ (*Figure 8D*). At no time during mechanical ventilation, or during the periods where rats were *briefly* removed from the ventilator, did active expiration occur in the absence of inspiration. As blood gases were normal (*Figure 8*), the loss of expiratory activity is unlikely to be due to excessive hypoxia or hypercapnia. Thus there appears to be a minimum level of respiratory network excitability required for active expiration. In the absence of preBötC activity, this network excitability may be provided by chemosensory drive, accounting for the $\int Abd_{EMG}$ activity seen shortly before the onset of apnea. However following recovery from apnea the increase in network excitability is provided by the preBötC either directly through its extensive excitatory projections throughout the respiratory network (*Tan et al., 2010*), e.g., the $pF_V$ that can modulate expiratory activity (*Marina et al., 2010*; *Huckstepp et al., 2015*), or indirectly when $V_T$ became large enough to induce mechanosensory feedback (that can provide expiratory drive, e.g., *Remmers, 1973*; *Davies and Roumy, 1986*; *Janczewski and Feldman, 2006*).

## Summary

We have developed a new strategy using complementary pharmacogenetics for studying coupled oscillator systems. By independently altering the excitability of two anatomically and functionally separate respiratory oscillators, we have uncovered a fundamental interaction and further delineated their role within the breathing CPG. We conclude that though respiration results from the interaction of two distinct oscillators, the preBötC (inspiration) and the $pF_L$ (expiration) are not organized as symmetrical half centers. Instead in the adult rat, the preBötC is the dominant oscillator and the $pF_L$ is the subsidiary conditional oscillator that is normally suppressed at rest; whereas the preBötC can drive breathing alone, the $pF_L$ is unable to drive breathing (including active expiration) in the absence of an additional form of network excitation, i.e., ongoing rhythmic preBötC activity sufficient to drive inspiratory motor output or increased chemosensory drive when changes in blood gases are below the threshold where hypoxia is sufficient to inhibit abdominal muscle recruitment. This hierarchy is established by the sensitivity of the system to each oscillator, even when preBötC drive is low, it is able to drive inspiratory bursts and inhibit expiration, whereas even when $pF_L$ drive is high, it requires a certain amount of network excitability from other sources to drive expiratory activity. This asymmetrical organization may be relevant to other neural networks that contain hierarchically organized coupled oscillators for pattern generation, such as may underlie the asymmetrical acts of flexion and extension in locomotion (*Grillner, 2003*; *Talpalar et al., 2013*). The interactions

of the preBötC and pF$_L$ change with development and maturation (*Figure 9*), and with state, represent another layer of complexity in understanding the neural control of breathing.

## Materials and methods

### Viral design and handling

Two different viruses were used: AAV-2/5 hSyn-HA-hM$_3$D(Gq)-IRES-mCitrine (6 $\times$ 10$^{12}$ vp/ml; HM$_3$DR; UNC Gene Therapy Vector Core, Chapel Hill, NC), and; AAV-DJ synapsin-allatostatin receptor-GFP (*Huckstepp et al., 2015*) (5.8 $\times$ 10$^{13}$ vp/ml; AlstR; SALK institute GT$^3$ Core, La Jolla, CA). The viruses were stored at -80°C. For injection, viruses were held at 4°C and loaded into pipettes.

### Viral transfection (*Figure 1*)

All protocols were approved by the UCLA Chancellor's Animal Research Committee. Male Sprague-Dawley rats (350–450 g) were anesthetized by intraperitoneal ketamine (100 mg/kg; Clipper Distribution Co, St Joseph, MO), xylazine (10 mg/kg; Lloyd Inc, Shenandoah, IA), and atropine (1 mg/kg; Westward Pharmaceutical Co, Eatontown, NJ), supplemented with isofluorane (0.5–2%; Piramal Healthcare Ltd, India) as required. Rats were placed prone in a stereotaxic apparatus (Kopf Instruments, St, Tujunga, CA), with the head positioned with Bregma 5 mm below Lambda, on a heating pad to maintain body temperature at 37 $\pm$ 0.5°C. The dorsal medullary surface was exposed and pipettes placed stereotaxically into the preBötC or pF$_L$. Coordinates were (lateral, rostral, ventral from obex in mm): preBötC; (2.0, 0.9, 2.8), and pF$_L$; (2.5, 1.8, 3.5). Viral solutions were pressure injected (100–150 nL) with a Picospritzer II (General Valve Corp, Fairfield, NJ) controlled by a pulse generator. Pipettes were left in place for 3–5 min to prevent solution backflow up the pipette track. We injected AlstR-expressing AAV in preBötC and HM$_3$DR-expressing AAV in pF$_L$. Postoperatively, rats received buprenorphine (0.1 mg/kg; Reckitt Benckiser, UK) intraperitoneally and meloxicam (2 mg/kg; Norbrook Inc, UK) subcutaneously, and antibiotics (10 days; TMS: Hi-Tech Pharmacal, Amityville, NY) and meloxicam (4 days; 0.05 mg/mL) in their drinking water. Rats were allowed 3–6 weeks for recovery and viral expression, with food and water *ad libitum*.

### Ventral approach

Anesthesia was induced with isofluorane and maintained with urethane (1.2–1.7 g/kg; Sigma, St Louis, MO) in sterile saline via a femoral catheter. Rats were placed supine in a stereotaxic apparatus on a heating pad to maintain body temperature at 37 $\pm$ 0.5°C. The trachea was cannulated, and respiratory flow was monitored via a flow head (GM Instruments, UK). A capnograph (Type 340: Harvard Apparatus, Holliston, MA) was connected to the tracheal tube to monitor expired $CO_2$, as a proxy of blood gases homeostasis. Paired electromyographic (EMG) wires (Cooner Wire Co, Chatsworth, CA) were inserted into genioglossal (GG), diaphragmatic (Dia), and oblique abdominal muscles (Abd). Anterior neck muscles were removed, a basiooccipital craniotomy exposed the ventral medullary surface, and the dura was resected. A bilateral vagotomy was performed to remove confounding factors such as feedback from lung stretch receptors that can drive abdominal activity (*Janczewski and Feldman, 2006*), after which exposed tissue around the neck and mylohyoid muscle were covered with dental putty (Reprosil; Dentsply Caulk, Milford, DE) to prevent drying. As micturition is inhibited under anesthesia, rats bladders were expressed preceding and during the experiment to remove any risk of autonomic dysreflexia from bladder distension; to maintain fluid balance rats were given an IP injection of saline every time the bladder was expressed. Rats were left for 30 min for breathing to stabilize. At rest, spontaneous breathing consisted of alternating active inspiration and passive expiration. From a ventral approach allatostatin (Alst; 10 μM; ~100-200 nL; Antagene Inc, Sunnyvale, CA) in sterile saline was injected bilaterally into the preBötC to hyperpolarize neurons transfected by AlstRs. Coordinates were (lateral from the basilar artery, caudal from the rostral hypoglossal nerve rootlet, dorsal from the ventral surface in mm): preBötC; (2.0, 0.6, 0.7). Small adjustments were made to avoid puncturing blood vessels. Rats were ventilated for the duration of ensuing apnea. After breathing stabilized, clozapine-N-oxide (CNO; 90 μM; Santa Cruz Biotechnology, Dallas, TX) in sterile saline was applied to the ventral medullary surface to depolarize pF$_L$ neurons transfected with HM$_3$DRs. Once breathing stabilized, rats received a second set of bilateral injections of Alst. Though rats were ventilated with room air for the duration of ensuing

apnea, they were intermittently removed from the ventilator to assess spontaneous breathing, and drives to inspiration and expiration. Ventilation depths and speed were chosen to match end-tidal $CO_2$ to that when the rat was spontaneously breathing room air. Once rats were spontaneously breathing and no longer required ventilation, CNO was removed from the ventral surface of the medulla and the medulla was washed in PBS. All 8 rats underwent the entire procedure, i.e., there were 8 biological replicates, and were only exposed to each condition once, i.e., there were no technical replicates. In age-matched rats not transfected with AlstR- or HM₃DR-expressing AAVs, we injected Alst into the preBötC or applied CNO to the medullary surface to see if these protocols produced non-specific effects All 8 rats underwent the entire procedure, i.e., there were 8 biological replicates, and were only exposed to each condition once, i.e., there were no technical replicates.

## Localization of transfected neurons

Rats were sacrificed by overdose of urethane and transcardially perfused with saline followed by cold (4°C) paraformaldehyde (PFA; 4%). The medulla was harvested and postfixed in 4% PFA overnight at 4°C, then cryoprotected in sucrose (30%) in standard PBS (1–3 days at 4°C). PBS contained (mM): NaCl 137, KCl 2.7, $Na_2HPO_4$ 10, $KH_2PO_4$ 1.8, pH 7.4. Brainstems were transversely sectioned at 40 μm. Free-floating sections were incubated overnight in PBS containing 0.1% Triton X-100 (PBT) and 1° antibodies (1:500): mouse anti-NeuN (EMD Millipore, Billerica, MA), rabbit anti-neurokinin 1 receptor (NK1R: EMD Millipore), and chicken anti-green fluorescent protein (GFP: Aves lab, Tigard, OR). The tissue was washed in PBS, 6 times for 5 min and then incubated separately for 2–4 hr in a PBT containing 2° antibodies (1:250): Donkey anti-mouse AlexFluor 647, donkey anti-rabbit rhodamine red, donkey anti-chicken AlexFluor 488 (Jackson ImmunoResearch Laboratories Inc, West Grove, PA). The tissue was washed in PBS, 6 times for 5 min. Slices were mounted onto polylysine-coated slides, dehydrated overnight at 22°C, and coverslipped using Cytoseal 60 (Electron Microscopy Sciences, Hatfield, PA). Slides were analyzed using a fluorescent microscope with AxioVision acquisition software (AxioCam2, Zeiss, Germany).

## Data analysis and statistics

Sample sizes were calculated using Gpower 3 v3.1.9.2 (http://www.ats.ucla.edu/stat/gpower/); using a 'means: Wilcoxon signed-rank test (matched pairs)' test, with a desired power of 90%, at a 5% significance level, and an effect size of 1.15 (calculated from the initial effect of Alst on respiratory frequency). Data were only included from animals where the preBötC and $pF_L$ were successfully targeted bilaterally, and no data were excluded from these animals. All statistical analysis was performed in Igor Pro (WaveMetrics, Lake Oswego, OR).

EMG signals and airflow measurements were collected using preamplifiers (P5; Grass technologies, Rockland, MA) connected to a Powerlab AD board (ADInstruments, Australia) in a computer running LabChart software (ADInstruments), and were sampled at 400 Hz/channel. High pass filtered (>0.1 Hz) flow head measurements were used to calculate: tidal volume ($V_T$, peak amplitude of the integrated airflow signal during inspiration, converted to mL by comparison to calibration with a 3 mL syringe), inspiratory duration ($T_I$, beginning of inspiration until peak $V_T$), expiratory duration ($T_E$, peak $V_T$ to the beginning of the next inspiration), and $f$ ($1/[T_I+T_E]$). EMG data, expressed in arbitrary units (A.U.), were integrated (τ=0.05 s; $\int Dia_{EMG}$, $\int GG_{EMG}$, and $\int Abd_{EMG}$) and peak amplitude of each signal was computed for each cycle. To obtain control values, all parameters except end-tidal $CO_2$ ($ETCO_2$), were averaged over 20 consecutive cycles preceding each experimental manipulation ($\overline{X}_{control}$). After Alst, measurements were taken at 2 time points: i) 20 cycles were averaged where only partial effects were seen, and; ii) 20 points were averaged following apnea. After CNO, 20 cycles were averaged after breathing had stabilized. After Alst in the presence of CNO, 20 points were averaged following apnea. In the presence or absence of CNO, capnograph peaks were averaged for 10 cycles preceding Alst, and for 5–10 cycles preceding apnea. Following removal from ventilation, the amplitude of the inspiratory bursts and $\int Dia_{EMG}$ and $\int GG_{EMG}$ activity immediately preceding the first $\int Abd_{EMG}$ burst (*Figure 8*, blue lines dashed lines) were recorded to calculate inspiratory parameters at which active expiration returned.

For each rat we obtained $\overline{X}_{control}$, and the average of 20 cycles during the stimulus ($\overline{X}_{stimulus}$). $\overline{X}_{control}$ values and their associated $\overline{X}_{stimulus}$ values for each parameter in each rat were combined into a single data set. To facilitate graphical comparisons data were normalized to the highest value in

the data set regardless of whether it belonged to $\overline{X}_{control}$ or $\overline{X}_{stimulus}$ group (C in *Figure 2*, *4*–*7*). Therefore the highest value in the data set, whether it be $\overline{X}_{control}$ or $\overline{X}_{stimulus}$, was 1.0; except for measurements of $V_T$, $\int Dia_{EMG}$, and $\int GG_{EMG}$ following ventilation, which are displayed as absolute values. Recorded data were not normally distributed, and were therefore analyzed using non-parametric statistical test, and reported as median and interquartile range (IQR). Statistical tests performed in Igor Pro (WaveMetrics), are 2-sided Wilcoxon signed-rank tests with a significance level of $p \leq 0.05$. Data are displayed as box and whisker plots for comparison of groups, and as line graphs for individual experiments. Kernel density estimations (*Parzen, 1962*; *Epanechnikov, 1969*), were used to determine the distribution of respiratory periods. After calculating the optimal bandwidth, i. e., bin size (*Park and Marron, 1990*; *Sheather and Jones, 1991*), the data was smoothed (*Cao et al., 1994*) and plotted. The modality of kernel density plots, were used to assess baseline respiratory periods and whether breathing slowed by quantal integers of that baseline. Bandwidth selection, data smoothing, and kernel density plots were performed in Microsoft excel (Microsoft Corporation, Redmond, WA) using an add-in written by the royal society of chemistry (http://www.rsc.org/Membership/Networking/InterestGroups/Analytical/AMC/Software/kerneldensities.asp).

## Additional information

### Funding

| Funder | Grant reference number | Author |
|---|---|---|
| National Institutes of Health | NIH HL 70029 | Jack L Feldman |

The funders had no role in study design, data collection and interpretation, or the decision to submit the work for publication.

### Author contributions

RTRH, Conception and design, Acquisition of data, Analysis and interpretation of data, Drafting or revising the article; LEH, KPC, Acquisition of data, Drafting or revising the article; JLF, Conception and design, Analysis and interpretation of data, Drafting or revising the article

### Author ORCIDs

Robert TR Huckstepp, http://orcid.org/0000-0003-4410-3397
Jack L Feldman, http://orcid.org/0000-0003-3692-9412

### Ethics

Animal experimentation: This study was performed in strict accordance with the recommendations in the Guide for the Care and Use of Laboratory Animals of the National Institutes of Health. All protocols (IACUC protocol #1994-159-73) were approved by the UCLA Chancellor's Animal Research Committee. Recovery surgery was performed under Ketamine/xylazine anesthesia and accompanied bybuprenorphine and meloxicam analgesia, non-recovery surgery was performed under urethane anesthesia, and every effort was made to minimize suffering.

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
