## [Decision Letter]

[Editors’ note: a previous version of this study was rejected after peer review, but the authors submitted for reconsideration. The first decision letter after peer review is shown below.]

Thank you for submitting your work entitled "Interactions between respiratory oscillators in adult rats" for consideration by *eLife*. Your article has been reviewed by three peer reviewers, one of whom is a member of our Board of Reviewing Editors, and the evaluation has been overseen by a Reviewing Editor and a Senior Editor. Our decision has been reached after consultation between the reviewers. Based on these discussions and the individual reviews below, we regret to inform you that your work will not be considered further for publication in *eLife*.

We recognize that this study describes the novel observation that active expiration turns tonic if the preBötzinger complex is experimentally silenced. The reviewers congratulate you and your co-authors for the cutting-edge approaches that allowed you to independently lesion the preBötzinger complex and pFRG region. Based on these manipulations, you and your co- authors conclude that active expiration (and the oscillator in the pFRG) requires rhythmic inhibition from the preBötC to generate expiration in the adult. Since this is not the case in juvenile animals, you proposed that there must be a developmental change in the role of the pFRG in breathing.

The decision to reject is based on the reviewers' major concern that these experiments do not suffice to justify your major conclusion. In all cases silencing the preBötzinger complex is associated with an apnea while the animal is not ventilated. Under these conditions it cannot be avoided that the animal will become hypoxic. Thus, it is not clear whether the cessation of inspiration and expiration is due to the cessation of the preBötzinger complex or the hypoxic conditions associated with the apnea. It is well-known that hypoxic conditions will lead to the cessation of inspiration and expiration. It is also well known that adults are more sensitive to hypoxia than juveniles/neonates.

Thus, the reviewers demand an additional set of new experiments. To justify your conclusion it is critical to silence the preBötC while ventilating the animal, show stable oxygenation in the absence of preBötC activity, and then demonstrate whether under these physiologically stable conditions active expiration can be induced, persists or disappears when inspiration ceases neuronally. These experiments will not be feasible within the allowed two months, hence we will not be able to consider your paper. For further details, please see the individual reviews.

*Reviewer #1:*

This study is an extension of a discovery that was made ten years ago (J.Physiol. 2006) indicating that two rhythm generators underlie the control of breathing: the preBötC for Inspiration, and the pFRG (first discovered by Onimaru) responsible for expiration ("active expiration"). The study does not contribute any new insights to this well-known concept, since this concept has been demonstrated not only in juveniles (the original 2006 study), but also in adults (J. Neuroscience, same lead author).

The only new discovery is based on the observation that active expiration turns tonic if the preBötzinger complex is lesioned/silenced. This observation leads the authors to conclude that active expiration (and the oscillator in the pFRG) requires rhythmic inhibition from the preBötC to generate expiration in the adult. The authors conclude, that there must be a developmental change in the role of the pFRG in breathing.

However, based on the experiments submitted here, this conclusion is not justified.

In all cases when apnea is induced by lesioning/silencing the preBötzinger complex the animal is not ventilated. This means, the animal will be exposed to severe hypoxia, which could (and will) indirectly induce the apnea and cessation of inspiration and expiration. This is a well-known phenomenon, and not at all surprising.

Thus, in order to arrive at the conclusion that the authors propose in this study, they need to add new experiments i.e. block or silence the preBötC while ventilating the animal show stable oxygenation and induce active expiration and then see whether under these physiologically stable conditions active expiration persists or disappears when inspiration ceases. Given that this is the major conclusion on which this paper rests, this experiment is absolutely essential.

Another major issue is the interpretation of the quantal slowing. First, it would be interesting to relate this to expiration, as has been done in prior studies. Secondly, the interpretation/speculation that the quantal slowing is explained by the presence of burstlets is not based on any experiment. These burstlets as far as I understand the original publication are "subthreshold" oscillations within the respiratory rhythm generator – without burst generation. If this is the case, how would the authors explain the transmission of these subthreshold oscillations to the motor output?

*Reviewer #2:*

I congratulate the investigators on a well-conducted study and a well-written manuscript that provides a significant advancement in understanding of mechanisms of respiratory rhythm generation. I am not requesting any specific changes in the manuscript prior to publication. However, I have a few suggestions for the authors to consider for additions to the discussion.

First, rostral pontine nuclei provide an important contribution to the control of breathing. What are the authors' speculations regarding this contribution to the two oscillators shown herein?

Second, what are the authors' thoughts on interactions between these two respiratory oscillators during normal physiological conditions particularly under conditions of NREM and REM sleep?

Third, this study shows the effects of acute, brief alterations in activity of the two oscillators. What happens when either one of the oscillators is altered on a long term or chronic basis? Studies by J. Wenninger et al., K. Krause et al., and S. Neumueller et al. seem to indicate that there is plasticity within the respiratory rhythm generation system.

*Reviewer #3:*

Authors investigated the interaction between two coupled oscillators in the lower brainstem of anesthetized adult rats. This is an important issue in neuroscience. The experimental design is straightforward and results are clear. However, I found some problems in the interpretation of results.

General comments:

The most important findings seemed to be that active expiration could not be induced when preBötC inspiratory driven activity was suppressed, based on results in Figure 7 and Figure 8. Indeed, Figure 7 (and also Figure 2) shows that cessation of rhythmic inspiratory activity after Alst injection in preBötC was followed by disappearance of Abd activity. Consequently, authors concluded that in adult rats active expiration is driven by the pF_L_ but requires ongoing preBötC activity. My concern is that these results may be due to transient unphysiological condition like extremely high CO_2_ (and low O_2_) of blood gases. To avoid such unphysiological situation, therefore, experiments should be done under following conditions to verify their hypothesis; using ventilator, CO_2_ level should be kept in suitable values to induce Abd activity in the presence (or absence) of CNO, then injection of Alst to preBötC should be performed to stop Dia activity and to examine Abd activity.

In Figure 3, they showed quantal inhibition of Dia activity. This result is very important, although they did not discuss much about this result. If possible, authors should also show Abd activity (or GG activity) together with Dia activity during quantal slowing. Analysis of such data may be useful to discuss interaction of two oscillators. The quantal inhibition of respiratory rhythm is known in newborn (and maybe juvenile) animal. Can this phenomenon in the present study be explained in similar mechanisms of interaction between two coupled oscillators like newborn rodent? It would be also helpful for further understanding of coupled oscillator to discuss whether there are any other examples of such quantal inhibition in adult animal.

Related to above questions, activity of two oscillators was assessed by motor neuron outputs in the present study. Central activity of oscillators may not always appear in motor output; i.e. motor output may be not equivalent to central activity to determine CPG property.

Specific comments:

Discussion section, "Shortly after birth (>1 day), the breathing central pattern generator (CPG) is driven by the preBötC (Oku et al., 2007) (Figure 9), and…”: This part is questionable. This optical recording paper seemed to show no direct evidence for postnatal change of the CPG and there is no electrophysiological study to support this idea. More accurate explanation is needed.

[Editors’ note: what now follows is the decision letter after the authors submitted for further consideration.]

Thank you for resubmitting your work entitled "Interactions between respiratory oscillators in adult rats" for further consideration at *eLife*. Your revised article has been favorably evaluated by a Senior editor and two reviewers, one of whom is a member of our Board of Reviewing Editors.

The manuscript has been improved but there are some remaining issues that need to be addressed before acceptance, as outlined below:

Reviewer #1:

This is a revision of a previously submitted manuscript. The authors have done a great job in addressing the reviewers’ comments. I have no further comments. But I think it is a solid contribution to the field, and the findings are novel and of general interest. The finding that the pFRG oscillator requires input from the preBötC is surprising which makes this paper very interesting.

*Reviewer #2:*

I know there were concerns previously that the absence of expiratory activity during prolonged silencing of the preBötC could be due to the effect of hypoxia on expiratory muscle activity. However the discussion in subsection “Respiratory network activity is a requirement for active expiration” seems to counter this concern. That is as the effect of hyperpolarizing effect of Alst on preBötC neurons progressively diminished, the rats were ventilated but upon removal from the ventilator, Abd activity did not return until presumably preBötC activity reached a threshold level. Dia and GG activity always returned before Abd activity. When the rats were ventilated to maintain normal blood gases, there was no Abd activity when the rats were briefly removed from the ventilator; thus, it does not seem that hypoxia could be causing the depressed Abd activity. Accordingly, I accept their point that a minimal amount of preBötC activity seems to be necessary drive the expiratory oscillator.

On the other hand, I am concerned by the conclusion as stated in the Abstract that "in anesthetized adult rats, active expiration is driven by the pF_L_ but requires an additional form of network excitation, i.e., ongoing rhythmic preBötC activity sufficient to drive inspiratory motor output or increased chemosensory drive". If increased chemosensory drive can drive the pF_L_ oscillator, then why doesn't the eventual severe hypercapnia during apnea produced by silencing the preBötC activate expiratory muscle activity? Am I missing something?

---

## [Author Response]

[Editors’ note: the author responses to the first round of peer review follow.]

We recognize that this study describes the novel observation that active expiration turns tonic if the preBötzinger complex is experimentally silenced. The reviewers congratulate you and your co-authors for the cutting-edge approaches that allowed you to independently lesion the preBötzinger complex and pFRG region. Based on these manipulations, you and your co- authors conclude that active expiration (and the oscillator in the pFRG) requires rhythmic inhibition from the preBötC to generate expiration in the adult. Since this is not the case in juvenile animals, you proposed that there must be a developmental change in the role of the pFRG in breathing.

The decision to reject is based on the reviewers' major concern that these experiments do not suffice to justify your major conclusion. In all cases silencing the preBötzinger complex is associated with an apnea while the animal is not ventilated. Under these conditions it cannot be avoided that the animal will become hypoxic. Thus, it is not clear whether the cessation of inspiration and expiration is due to the cessation of the preBötzinger complex or the hypoxic conditions associated with the apnea. It is well-known that hypoxic conditions will lead to the cessation of inspiration and expiration. It is also well known that adults are more sensitive to hypoxia than juveniles/neonates.

Thus, the reviewers demand an additional set of new experiments. To justify your conclusion it is critical to silence the preBötC while ventilating the animal, show stable oxygenation in the absence of preBötC activity, and then demonstrate whether under these physiologically stable conditions active expiration can be induced, persists or disappears when inspiration ceases neuronally. These experiments will not be feasible within the allowed two months, hence we will not be able to consider your paper. For further details, please see the individual reviews.

Please accept the resubmission our manuscript for eLife. In light of the reviewers comments we have performed additional analysis of the data to support our findings.

We have provided an addition panel to Figure 8 from rats we ventilated in the presence of CNO (Pink shading) where end-tidal CO_2_ was maintained at normal resting levels (~45 mmHg). This panel shows that active expiration could occur during ventilation, and that no expiratory activity was seen until inspiratory activity returned during ventilation.

Furthermore, upon revising the manuscript we realized that certain statements made within the text could have been made clearer; for example that in the absence of preBӧtC activity, active expiration can occur in the presence of increased chemosensory drive. We recognize that as authors we must accept the blame when readers, including reviewers, miss key points. We have rewritten the manuscript to make this and other points even more explicit.

Reviewer #1:

This study is an extension of a discovery that was made ten years ago (J.Physiol. 2006) indicating that two rhythm generators underlie the control of breathing: the preBötC for Inspiration, and the pFRG (first discovered by Onimaru) responsible for expiration ("active expiration"). The study does not contribute any new insights to this well-known concept, since this concept has been demonstrated not only in juveniles (the original 2006 study), but also in adults (J. Neuroscience, same lead author).

We appreciate the reviewer’s statement that our proposal that there are separate inspiratory and expiratory oscillators is a “well-known” concept, but there is much work to be done to show how these oscillators interact, and there are still those not convinced of this. We are unaware of any paper (ours or anyone else’s) that looks at active expiration after silencing the preBötC, or how inspiratory and expiratory oscillators interact to pattern breathing in the adult rat. Thus this study does indeed contribute several key new insights to this “well-known” concept.

The only new discovery is based on the observation that active expiration turns tonic if the preBötzinger complex is lesioned/silenced. This observation leads the authors to conclude that active expiration (and the oscillator in the pFRG) requires rhythmic inhibition from the preBötC to generate expiration in the adult. The authors conclude, that there must be a developmental change in the role of the pFRG in breathing.

This reviewer’s surmise of our conclusion is not what we intended to convey; our fault for not making our conclusion sufficiently transparent. We did not say that active expiration requires inhibitory input; rather we said that the phasic pattern of expiration requires inhibition from the preBötC. This is based on the observation that when the preBötC stops firing (including preBötC inhibitory neurons) then the expiratory rhythm transforms from phasic to tonic; this is an idea first proposed by Sears and colleagues. We stated in the Discussion section “We suggest, like Sears et al., that near the transition from eupnea to apnea, expiratory activity is phasic due to periodic inhibition of tonic activity during inspiration, and becomes tonic once the phasic inspiratory inhibition is lost (Figure 2, Figure 7, Figure 9). We suggest the loss of phasic inhibition is the result of silencing of inhibitory preBötC inspiratory neurons by Alst (Kuwana et al., 2006; Morgado-Valle et al., 2010), implying a significant role for inhibitory preBötC neurons in shaping expiratory output.”

Moreover, we actually postulated that silencing of inhibitory neurons in the preBӧtC may actually increase expiratory activity, through disinhibition of the pF_L_. As we stated “Alternatively, active expiration could be due to disinhibition of a conditional expiratory oscillator resulting from a loss of (presumptive) inhibitory preBötC drive (Kuwana et al., 2006; Morgado-Valle et al., 2010) to the pF_L_ (Figure 9).”

However, based on the experiments submitted here, this conclusion is not justified.

In all cases when apnea is induced by lesioning/silencing the preBötzinger complex the animal is not ventilated. This means, the animal will be exposed to severe hypoxia, which could (and will) indirectly induce the apnea and cessation of inspiration and expiration. This is a well-known phenomenon, and not at all surprising.

We explicitly acknowledged the inhibitory effect of hypoxia, and its role in the cessation of on expiratory activity. As we stated in the Discussion section “when inspiratory motor outflow disappeared and hypercapnia and hypoxia inexorably increased, active expiration ceased (Figure 2, Figure 9, see also Tan et al., 2008), perhaps due to the inhibitory effect of severe hypoxia on expiratory motor output (Sears et al., 1982; Fregosi et al., 1987)”.

In fact, it is most likely the hypercapnia and hypoxia during hypopnea that allows tonic activity of abdominal muscles to continue after inspiratory activity ceases. As seen in Figure 2, where there is expiratory activity without activation of HM_3_DRs in the pF_L_, and stated in the Discussion section “In either case, active expiration could have been due to increased hypoxic and/or hypercapnic drive to the pF_L_ as inspiratory movements waned”

Thus, in order to arrive at the conclusion that the authors propose in this study, they need to add new experiments. I.e. block or silence the preBötC while ventilating the animal show stable oxygenation and induce active expiration and then see whether under these physiologically stable conditions active expiration persists or disappears when inspiration ceases. Given that this is the major conclusion on which this paper rests, this experiment is absolutely essential.

Even though we acknowledge the role of hypoxia on the cessation of expiratory activity, a careful examination of the data and a conservative interpretation leads to the parsimonious conclusion that hypoxia per se could NOT have participated in the absence of expiration activity during ventilation, or the delay in expiratory activity following removal from the ventilator. Rats were ventilated on average for ~18 min following the onset apnea after which removal from the ventilator led to the onset of inspiratory activity (Figure 8). Thus, rats were normoxic for ~18 min during which time the pF_L_ was being excited by the activation of HM_3_DRs and the preBötC was inhibited by activation of AlstRs and presumably silent; during this time, no abdominal activity was seen. That expiratory activity returned within 30 secs of being removed from the ventilator (Figure 8) means that the HM_3_DRs were still driving the pF_L_, and the pF_L_ was still capable of driving abdominal activity. The lack of abdominal activity during ventilation was not the result of mechanical actions of the ventilator, as expiratory activity was apparent during ventilation towards the end of the experiment (see Figure 8). Therefore, the most parsimonious explanation for the lack of abdominal activity during ventilation when inspiratory motor activity is absent is that expiratory drive from the pF_L_ alone is insufficient to drive expiration and requires an additional source of network excitation, such as inspiratory activity or chemosensory drive.

We have added a new panel in Figure 8, panel D, to show that expiration returns after inspiratory activity, and that abdominal activity can still occur during ventilation. We have also expanded the Results, Discussion and Summary to make this clearer.

Results section”: “Rats were mechanically ventilated to restore CO_2_ and O_2_ levels to within the normal range (Figure 7). […] Therefore, following silencing of the preBötC with Alst in the presence of CNO to drive the pF_L_ at no time during the re-initiation of spontaneous breathing, either with or without mechanical ventilation, did active expiration occur in the absence of inspiratory activity and chemosensory drive.”

Discussion section: “Following apnea during ventilation to maintain blood gases, when pF_L_neurons were excited by activation of HM_3_DRs there were three phases to the re-initiation of breathing. […] As blood gases were normal (Figure 8), the loss of expiratory activity is unlikely to be due to excessive hypoxia or hypercapnia.”

Summary: “the pF_L_ is unable to drive breathing (including active expiration) in the absence of an additional form of network excitation, i.e., preBötC activity or chemosensory drive. This hierarchy is established by the sensitivity of the system to each oscillator, even when preBӧtC drive is low, it is able to drive inspiratory bursts and inhibit expiration, whereas even when pF_L_ drive is high, it requires a certain amount of network excitability from other sources to drive expiratory activity.”

It was in fact essential to the conclusions of this paper that rats were allowed to become moderately hypercapnic and moderately hypoxic during the hypopnea leading up to the apnea, as this allowed us to see that in the absence of inspiratory activity, abdominal activity could persist if network excitability was increased by chemosensory drive, if only for a short period until it was inhibited by severe hypoxia. Furthermore, following apnea rats were immediately placed on the ventilator to remove hypoxic inhibition of expiratory activity, which showed that in conditions with no preBötC activity and no hypercapnic drive in normoxic conditions there was no expiratory activity. The experimental design allowed us to assess expiratory activity under the following conditions.

pF_L_ activitypreBӧtC activityHypercapniaHypoxiaAbdominal activityNoYesNoNoAbsentYes (driven by CO_2_)LowModerateModeratePhasicYes (driven by CO_2_)AbsentModerateModerateTonicNo (inhibited by hypoxia)AbsentHighHighAbsentNoAbsentNoNoAbsentYes (driven by HM_3_DR)YesNoNoPhasicYes (driven by HM_3_DR and CO_2_)LowModerateModeratePhasicYes (driven by HM_3_DR and CO_2_)AbsentModerateModerateTonicYes (driven by HM_3_DR, but inhibited by hypoxia)AbsentHighHighAbsentYes (driven by HM_3_DR)AbsentNoNoAbsent

We have also expanded the Methods section to make our protocol clearer. “Though rats were ventilated with room air for the duration of ensuing apnea, they were intermittently removed from the ventilator to assess spontaneous breathing, and drives to inspiration and expiration. […] Once rats were spontaneously breathing and no longer required ventilation, CNO was removed from the ventral surface of the medulla and the medulla was washed in PBS.”

Another major issue is the interpretation of the quantal slowing. First, it would be interesting to relate this to expiration, as has been done in prior studies. Secondly, the interpretation/speculation that the quantal slowing is explained by the presence of burstlets is not based on any experiment. These burstlets as far as I understand the original publication are "subthreshold" oscillations within the respiratory rhythm generator – without burst generation. If this is the case, how would the authors explain the transmission of these subthreshold oscillations to the motor output?

In our original burstlet paper (Kam et al. 2013), burstlets are by definition subthreshold events. However, as made clear in Kam et al. 2013 paper (Figure 7 and the associated text), burstlets can be transmitted to motor output if specific conditions are met; namely lowered excitability in preBötC with increased drive to respiratory motor pools. We believe that these conditions are met in our experiments: lowered drive in the preBötC resulting from activation of the allatostatin receptor depresses bursting while burstlets may continue in the preBötC. Under normal conditions burstlets are not strong enough to drive activity in motor pools. However, increased excitability in the motor pools caused by increased chemosensory drive (from decreased ventilation) causes these normally subthreshold events to become suprathreshold and thus these burstlets are transmitted to the motor pool. As we stated in the Results section, “This low amplitude Dia_EMG_ and airflow activity are postulated to represent the inspiratory motor outflow, i.e., EMG, manifestation of preBötC burstlets (Kam et al., 2013), i.e., low levels of rhythmogenic neural population activity in the preBötC that under normal conditions occur in the absence of motor output, being transmitted to motoneuron pools (see Figure 7 in Kam et al., 2013).”

We have also expanded the Discussion to make this clearer. “Here, lowered drive in the preBötC from activation of the allatostatin receptor creates burstlets in the preBötC; under normal conditions these burstlet signals are not strong enough to drive motor activity. […] Interestingly, even when expiratory activity was at its highest, low level preBötC activity, as seen as burstlets on the Dia_EMG_ and GG_EMG_ was still able to inhibit Abd_EMG_ activity (Figure 7).”

Reviewer #2:

I congratulate the investigators on a well-conducted study and a well-written manuscript that provides a significant advancement in understanding of mechanisms of respiratory rhythm generation. I am not requesting any specific changes in the manuscript prior to publication. However, I have a few suggestions for the authors to consider for additions to the discussion.

First, rostral pontine nuclei provide an important contribution to the control of breathing. What are the authors' speculations regarding this contribution to the two oscillators shown herein?

Second, what are the authors' thoughts on interactions between these two respiratory oscillators during normal physiological conditions particularly under conditions of NREM and REM sleep?

Third, this study shows the effects of acute, brief alterations in activity of the two oscillators. What happens when either one of the oscillators is altered on a long term or chronic basis? Studies by J. Wenninger et al., K. Krause et al., and S. Neumueller et al. seem to indicate that there is plasticity within the respiratory rhythm generation system.

We are delighted that this reviewer enjoyed our manuscript. Although no specific changes were requested, we did consider the reviewer’s suggestions in the revised manuscript.

Though the contribution of respiratory-related areas to network output is beyond the remits of this study, we do not in any way rule it out. In fact some of the chemosensory drive seen during moderate hypercapnia and hypoxia likely comes from sources outside of the preBötC and pF_L_. So as not to break up the flow of the discussion we have added a section to the technical considerations section to discuss this possibility. Results section “Here we discuss the output of the preBötC and pF_L_, which ultimately form the final output of the respiratory network. […]Therefore, the activity of the preBötC and pF_L,_ was assessed by motor output recorded from respiratory muscles, which may not always reflect the activity of these oscillators, as they may have subthreshold activity, e.g., Kam et al., 2013.”

In adult rats, there is no expiratory activity seen during sleep under normal physiological conditions, so we expect the interaction between the oscillators to be the same during sleep as they are during quiet wakefulness. To make this clear we have slightly expanded the line in the Introduction section to read: “In mammals at rest, during wakefulness and sleep, when active breathing movements are primarily inspiratory, generation of the underlying rhythm appears driven by the preBötC”.

The effect of chronic lesioning of the respiratory oscillators and network compensation for these changes is complicated. The Neumueller et al. 2011 paper, shows that destruction of the preBӧtC can be compensated for if the lesion occurs incrementally over time, however the Tupal et al. 2014 paper shows that loss of DBX1 neurons from birth cannot be compensated for. To properly address this subject would take considerable discussion in the manuscript, thus to keep our discussion concise we have not covered this subject in this paper.

Reviewer #3:

Authors investigated the interaction between two coupled oscillators in the lower brainstem of anesthetized adult rats. This is an important issue in neuroscience. The experimental design is straightforward and results are clear. However, I found some problems in the interpretation of results.

General comments:

The most important findings seemed to be that active expiration could not be induced when preBötC inspiratory driven activity was suppressed, based on results in Figure 7 and Figure 8. Indeed, Figure 7 (and also Figure 2) shows that cessation of rhythmic inspiratory activity after Alst injection in preBötC was followed by disappearance of Abd activity. Consequently, authors concluded that in adult rats active expiration is driven by the pF_L_ but requires ongoing preBötC activity.

This is partly correct, we do conclude that preBӧtC is a requirement for rats who are normoxic and normocapnic, hence the return of inspiratory activity always precedes the return of expiratory activity following apnea and ventilation. However, we conclude that expiratory activity can occur in the absence of inspiration if chemosensory drive is high, hence the appearance of tonic abdominal activity following the cessation of inspiratory activity. We concluded that expiratory activity requires a certain level of network activity from another source, either inspiratory activity or chemosensory drive. As we stated “Thus there appears to be a minimum level of respiratory network excitability required for active expiration. […] However following recovery from apnea the increase in network excitability is provided by the preBötC either directly through its extensive excitatory projections throughout the respiratory network (Tan et al., 2010), e.g., the pF_V_ that can modulate to expiratory activity (Marina et al., 2010; Huckstepp et al., 2015), or indirectly when V_T_ became large enough to induce mechanosensory feedback (that can provide expiratory drive (Remmers, 1973; Davies and Roumy, 1986; Janczewski and Feldman, 2006))”

We have also expanded the Abstract, Introduction, Discussion and Summary to make this clearer.

Introduction: “We conclude that in anesthetized adult rats active expiration is driven by the pF_L_ but requires an additional form of network excitation, i.e., ongoing rhythmic preBötC activity sufficient to drive inspiratory motor output or increased chemosensory drive.”

Introduction: “Importantly, we found that active expiration could not be induced when preBötC inspiratory driven motor activity was suppressed and chemosensory drive was absent, indicating that in adult rats active expiration is driven by the pF_L_ but requires an additional source of network excitation such as ongoing preBötC activity or chemosensory drive.”

Discussion section: “The pF_L_ is the subsidiary conditional oscillator, quiescent at rest (Figure 9); in the presence of concurrent, or reduced, preBötC activity (Figure 2), and appropriate inputs, such as an increase in expiratory drive through activation of exogenous receptors (Figure 6, see also Pagliardini et al., 2011) or altered blood gases (Huckstepp et al., 2015), the pF_L_ drives active expiration (Figure 9). In the absence of preBötC activity, chemosensory drive increases network excitability and drives active expiration, though abdominal activity is tonic due to a lack of phasic inhibition from the preBötC (Figure 9).”

Summary: “the pF_L_ is unable to drive breathing (including active expiration) in the absence of an additional form of network excitation, i.e., preBötC activity or chemosensory drive. This hierarchy is established by the sensitivity of the system to each oscillator, even when preBӧtC drive is low, it is able to drive inspiratory bursts and inhibit expiration, whereas even when pF_L_ drive is high, it requires a certain amount of network excitability from other sources to drive expiratory activity.”

My concern is that these results may be due to transient unphysiological condition like extremely high CO_2_ (and low O_2_) of blood gases. To avoid such unphysiological situation, therefore, experiments should be done under following conditions to verify their hypothesis; using ventilator, CO_2_ level should be kept in suitable values to induce Abd activity in the presence (or absence) of CNO, then injection of Alst to preBötC should be performed to stop Dia activity and to examine Abd activity.

For CO_2_ levels to be kept in suitable values to induce Abd activity in the absence of CNO, the rat would have to be chronically hypercapnic. In that situation, we would expect abdominal activity to continue, as chemosensory drive can drive abdominal activity in the absence of inspiratory activity, see previous answer. As discussed above, our experimental design allowed us to test the interactions of the preBötC and pF_L_ under an array of conditions. See table above.

In Figure 3, they showed quantal inhibition of Dia activity. This result is very important, although they did not discuss much about this result. If possible, authors should also show Abd activity (or GG activity) together with Dia activity during quantal slowing. Analysis of such data may be useful to discuss interaction of two oscillators. The quantal inhibition of respiratory rhythm is known in newborn (and maybe juvenile) animal. Can this phenomenon in the present study be explained in similar mechanisms of interaction between two coupled oscillators like newborn rodent? It would be also helpful for further understanding of coupled oscillator to discuss whether there are any other examples of such quantal inhibition in adult animal?

The effect of quantal slowing on ABD and GG in conjunction with Dia can be seen in Figure 2 and Figure 7, and so was not included in Figure 3, which shows burstlet activity and how quantal slowing was calculated. We have added the following sentence to expand on this in the Discussion, “During quantal slowing of breathing, inspiratory activity on the Dia_EMG_ and GG_EMG_ was interspersed with Abd_EMG_ activity (Figure 2, Figure 7). Interestingly, even when expiratory activity was at its highest, low level preBötC activity, as seen as burstlets on the Dia_EMG_ and GG_EMG_ was still able to inhibit Abd_EMG_ activity (Figure 7).”

Related to above questions, activity of two oscillators was assessed by motor neuron outputs in the present study. Central activity of oscillators may not always appear in motor output; i.e. motor output may be not equivalent to central activity to determine CPG property.

We agree that motor output may not always reflect oscillator activity, as described in our burstlet paper (Kam et al., 2013). We added the following statement to the technical considerations section to make this clear. “In addition, due to the necessity to perform injections into the medulla, we were unable to record directly from the preBötC or pF_L_. Therefore, the activity of the preBötC and pF_L,_ was assessed by motor output recorded from respiratory muscles, which may not always reflect the activity of these oscillators, as they may have subthreshold activity, e.g., Kam et al., 2013.”

Specific comments:

Discussion section, "Shortly after birth (>1 day), the breathing central pattern generator (CPG) is driven by the preBötC (Oku et al., 2007) (Figure 9), and…”: This part is questionable. This optical recording paper seemed to show no direct evidence for postnatal change of the CPG and there is no electrophysiological study to support this idea. More accurate explanation is needed.

We have added two new references, which provide evidence of the maturation of respiratory system from an ePF/pFRG lead network to a preBӧtC lead network.

Discussion section: “Shortly after birth (~P2-P4), the breathing central pattern generator (CPG) matures, and becomes driven by the preBötC (Oku et al., 2007; Mckay et al., 2009; Kennedy, 2015) (Figure 9), and remains so throughout life.”

[Editors' note: the author responses to the re-review follow.]

The manuscript has been improved but there are some remaining issues that need to be addressed before acceptance, as outlined below:

Reviewer #1:

This is a revision of a previously submitted manuscript. The authors have done a great job in addressing the reviewers comments. I have no further comments. But I think it is a solid contribution to the field, and the findings are novel and of general interest. The finding that the pFRG oscillator requires input from the preBötC is surprising which makes this paper very interesting.

We thank the reviewer for his/her kind comments.

Reviewer #2:

I know there were concerns previously that the absence of expiratory activity during prolonged silencing of the preBötC could be due to the effect of hypoxia on expiratory muscle activity. However the discussion in subsection “Respiratory network activity is a requirement for active expiration” seems to counter this concern. That is as the effect of hyperpolarizing effect of Alst on preBötC neurons progressively diminished, the rats were ventilated but upon removal from the ventilator, Abd activity did not return until presumably preBötC activity reached a threshold level. Dia and GG activity always returned before Abd activity. When the rats were ventilated to maintain normal blood gases, there was no Abd activity when the rats were briefly removed from the ventilator; thus, it does not seem that hypoxia could be causing the depressed Abd activity. Accordingly, I accept their point that a minimal amount of preBötC activity seems to be necessary drive the expiratory oscillator.

We are pleased that our revisions to the manuscript have made this point clearer.

On the other hand, I am concerned by the conclusion as stated in the Abstract that "in anesthetized adult rats, active expiration is driven by the pF_L_ but requires an additional form of network excitation, i.e., ongoing rhythmic preBötC activity sufficient to drive inspiratory motor output or increased chemosensory drive". If increased chemosensory drive can drive the pF_L_ oscillator, then why doesn't the eventual severe hypercapnia during apnea produced by silencing the preBötC activate expiratory muscle activity? Am I missing something?

We previously showed that both hypercapnia and hypoxia, when moderate and brief, lead to expiratory-related abdominal activity (Huckstepp et al., J Neurosci, 2016). In the paper under review, we conclude that the moderate hypercapnia and hypoxia following cessation of inspiratory activity is responsible for the tonic abdominal muscle activity at the beginning of apnea. As stated in the Discussion section “In either case, active expiration could have been due to increased hypoxic and/or hypercapnic drive to the pF_L_ as inspiratory movements waned”. This underlies our statement in the Summary that additional network excitation from increased chemosensory drive can lead to abdominal activity in the absence of preBötC activity.

We acknowledge that severe hypoxia profoundly inhibits abdominal muscle recruitment (Fregosi et al., J Appl Physiol, 1987), suggesting that the cessation of abdominal activity during apnea is due to severe hypoxia resulting from the prolongation of the apnea. We state: “when inspiratory motor outflow disappeared and hypercapnia and hypoxia inexorably increased, active expiration ceased (Figure 2, Figure 9, see also Tan et al., 2008), perhaps due to the inhibitory effect of severe hypoxia on expiratory motor output (Sears et al., 1982; Fregosi et al., 1987)”.

Nonetheless, we maintain that a detailed examination of the data leads to the parsimonious conclusion that hypoxia does NOT underlie the absence of expiratory activity during ventilation (that assures proper gas exchange and avoids hypoxia and hypercapnia), or the delay in the onset of expiratory activity following removal from the ventilator (see the reviewer’s previous comment and section “Respiratory network activity is a requirement for active expiration for discussion”).

In our Summary and Abstract we state that an additional form of network excitation is required for active expiration in anesthetized adult rats, as even when driving the pF_L_ by activation of the HM_3_D receptor, abdominal activity does not occur in the absence of another source of network excitation. We conclude that the additional network excitation necessary for recruitment of abdominal muscles can come from: 1) ongoing rhythmic preBötC activity sufficient to drive inspiratory motor output, as abdominal activity is not seen in the absence of inspiratory motor output when chemosensory drive is low or absent, or 2) increased chemosensory drive, as tonic abdominal activity is seen immediately after the cessation of inspiratory activity, when drive from the preBötC is low or absent but chemosensory drive is moderate.

We have redacted the Summary to make this point clearer: “the pF_L_is unable to drive breathing (including active expiration) in the absence of additional network excitation, i.e., ongoing rhythmic preBötC activity sufficient to drive inspiratory motor output or increased chemosensory drive when changes in blood gases are below the threshold where hypoxia is sufficient to inhibit abdominal muscle recruitment.”